# QUANTUM NEURAL FIELDS

## ABSTRACT

This paper introduces a new type of neural field for visual computing with components compatible with gate-based quantum hardware or simulators thereof. Our Quantum Neural Field Network (QNF-Net) expects as input a query coordinate and, optionally, a latent variable value, and outputs the corresponding field value. QNF-Net includes a new feature map for classical data encoding and a parametrised quantum circuit. The proposed neuro-deterministic data encoding converts, into qubit amplitudes, an energy spectrum of the Gibbs-Boltzmann distribution corresponding to the learned problem energy manifold. We provide a theoretical analysis of the model and its components and perform experiments on a simulator of a gate-based quantum computer with 2D images and 3D shapes (and their collections as learnt priors) and compare results with several classical baselines. QNF-Net consistently outperforms the classical baselines with a comparable number of parameters and achieves faster convergence speed, therefore showing its potential quantum advantages, even for relatively large-scale problems compared to what has been demonstrated in quantum machine learning so far. We will release the source code to facilitate method reproducibility.

## 1 INTRODUCTION

Coordinate-based neural fields are at the cornerstone of scene representation learning and they are widely used and indispensable in modern computer vision Xie et al. (2022a); Mildenhall et al. (2021); Park et al. (2019); Sabella (1988); Shue et al. (2023); Feng et al. (2022); Osher & Fedkiw (2005). They find applications in robotics Wiesmann et al. (2023); Maggio et al. (2023); Kwon et al. (2023), 3D reconstruction Williams et al. (2022); Sitzmann et al. (2021); Sun et al. (2022); Zhang et al. (2021); Ran et al. (2023) and novel view synthesis Ye et al. (2023); Li et al. (2021b;a); Mildenhall et al. (2021), to name a few areas. Neural fields are often used to continuously parameterise 2D images or 3D scenes and they encode various characteristics of a scene (such as 3D geometry, appearance, and material properties Yang et al. (2021); Shue et al. (2023); Courant et al. (2023)); they provide data priors to other methods and allow scene manipulation and editing (such as interpolation in the latent space, scene inpainting or completion Mirzaei et al. (2023)). The prominent advantages of neural fields include support of different scene topologies and a wide range of scene resolutions as well as balancing data fitting and generalisation. All these applications became possible in recent years, as there has been a notable shift from hand-crafted priors, primarily based on heuristics, to learning priors in the form of neural fields directly from data Xie et al. (2022b), with multi-layer perceptron (MLP) with ReLU activation being one popular building block for such a neural field, in the early days. Highly desirable characteristics of neural scene representations such as efficient and fast training, lightweightness and high accuracy, however, are still not easily combinable using modern neural approaches. Moreover, training neural fields can be computationally and resource-demanding, depending on the model and the data collection size. Hence, *any possible (even seemingly small) reduction in the required number of parameters and training iterations would be advantageous in widespread techniques and applications relying on neural fields.*

With emerging interest in variational quantum circuits, i.e. quantum machine learning (QML) Schuld et al. (2015); Biamonte et al. (2017); Cerezo et al. (2022) and given that quantum machine learning (QML) operate under fundamentally different principles compared to its classical counterpart, we shift our focus to QML and hope it can address those open challenges mentioned above. QML, which can execute on gate-based quantum hardware or simulators thereof, takes advantage of the gate-based quantum computational paradigm, i.e. the associated quantum-mechanical effects such

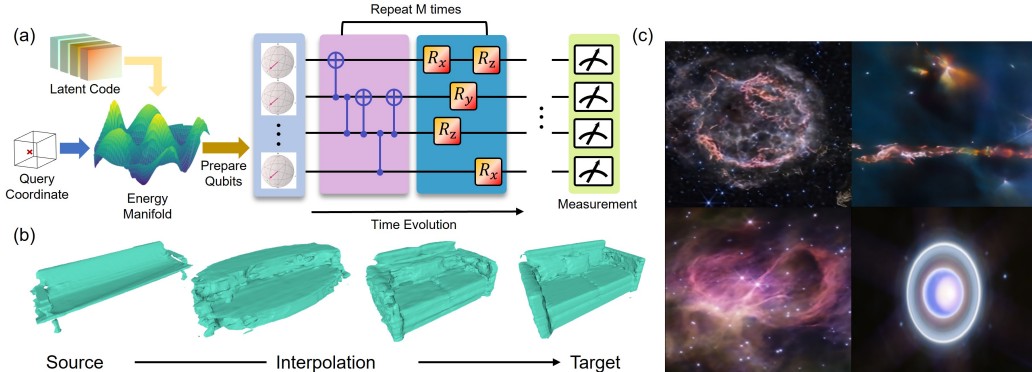

Figure 1: Our coordinate-based Quantum Neural Field Network (QNF-Net) can be learnt from data and can represent various signals: (a:) Overall architecture scheme; (b:) Latent space interpolation of 3D shapes; (c:) 2D images of high resolutions ($400 \times 350$ pixels) Gardner (2022).

as qubit superposition and entanglement. This provides an alternative to the classical neural universal approximators and could provide characteristics possibly not observed or easily achievable by classical architectures (such as faster model convergence and fewer parameters; learning different patterns compared to classical models). However, it is an open question how these postulated advantages translate to practical QML implementations. We, thus, introduce QNF-Net, a new architecture for learning Quantum Neural Fields (QNF) (see Figs. 1 and 2). QNF-Net is the *first quantum scene representation of its kind* that encodes a scene using parametrised quantum circuits (PQCs) learnt from data. More specifically, applying PQCs for neural field learning has multiple reasons and advantages that we observe empirically on a simulator of a fault-tolerant quantum computer. First, it results in faster convergence and fewer parameters to reach performance on par or better than classical methods. Second, PQCs allow learning patterns from data not accessible to classical models. Since PQCs can be interpreted as truncated Fourier series with coefficients determined through unitary quantum operations, they support improved and faster learning of high-frequency details in the input data. Instead of using heuristic encoding as most other work do Weigold et al. (2020); Huang et al. (2018); Schalkers & Möller (2024); Bondarenko & Feldmann (2020); Rathi et al. (2023)—which lacks generality and depends on specific applications—we also provide a learnable encoding strategy which projects classical data into quantum states with theoretical analysis. Moreover, efficiently encoding signals into quantum states remains challenging in QML. In this regard, QNF-Net could also be regarded as a novel and efficient way to encode data in learnable parameters of quantum circuits. To summarise, the primary technical contributions of this paper are as follows:

- QNF-Net, i.e. a hybrid quantum neural architecture for quantum neural field (QNF) learning that can be trained and run on simulators of fault-tolerant quantum hardware and, potentially, upcoming gate-based quantum machines (Sec. 3);
- Neuro-deterministic encoding of classical data through amplitude encoding of state probabilities derived from the inferred problem energy manifold (Sec. 3.1);
- Effective and efficient (in terms of the number of parameters) quantum circuit with a theoretical mathematical analysis of its expressiveness (Sec. 3.2);
- QNF-Net conditioning on a latent variable, which enables multiple applications such as shape interpolation in the latent space (Sec. 3.3).

QNF is a new way of representing images, 3D shapes and their collections on quantum hardware.

## 2 RELATED WORK

**Classical Neural 2D/3D Scene Representation.** Neural networks have been extensively used for implicitly learning scene representations Molaei et al. (2023); Chen et al. (2017); Tschernezki et al. (2022); Chen et al. (2021); Li et al. (2022). Moreover, the past few years have witnessed significant

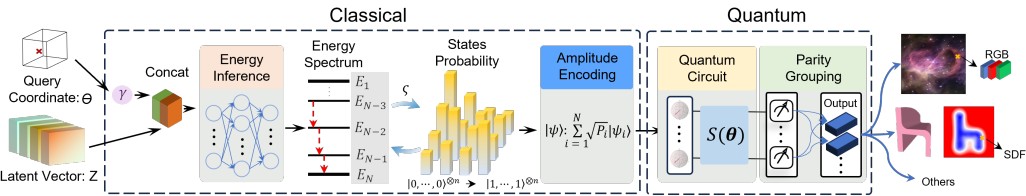

Figure 2: **Overview of the proposed QNF-Net, a hybrid quantum framework for representing field information (e.g., 2D images and 3D shapes).** The scene coordinates $\theta$ encoded using $\gamma$ (positional encoding) concatenated with the conditioning latent code $\mathbf{z}$ are used to infer the energy spectrum $\mathbf{E}$ of a quantum system, associated with statistical uncertainty $\mathbf{P}$ modelled by Boltzmann distribution $\varsigma$. This inferred statistical property is then processed by a parametrised quantum circuit $S(\theta)$ followed by qubit measurements. The measured values are grouped using the parity mapper to ensure consistent output dimensions.

progress, especially on 3D representations such as SDF Park et al. (2019); Duan et al. (2020), multi-view images Su et al. (2015); Yang et al. (2018), 3D shape manifold meshes Graham & Van der Maaten (2017); Graham et al. (2018), radiance fields Mildenhall et al. (2021) and graphs Wang et al. (2019). Among these, DeepSDFs and similar approaches Park et al. (2019); Mescheder et al. (2019) encode signed distance fields implicitly capturing the scene geometry. DeepSDFs can be conditioned on a latent variable that would allow learning shape collections in a single neural field and adjusting the decoded shapes at test time. In contrast to DeepSDF, our method is a QML approach, i.e., the SDFs of 3D shapes are encoded in parametrised quantum circuits. While QNF-Net preserves the core functionality of DeepSDF (e.g., support of topological changes), we reduce the training time and the overall number of parameters compared to it. QNF-Net can leverage gate-based quantum hardware which can be more energy-efficient or computationally faster for specific problems.

**Gate-based Quantum Computer Vision/Computer Graphics (QCV/CG).** This emerging inter-disciplinary field at the intersection of 3D vision, computer graphics, and quantum computing attracts growing attention. A few studies explored the application of quantum machine learning (QML) in QCV/CG. Early works Shiba et al. (2019) introduced a quantum circuit model for image denoising. They drew inspiration from convolutional principles, considering only interactions between each pixel and its neighbors to minimize qubit usage. Similarly, Cong et al. (2019) proposed quantum convolutional neural networks (QCNNs), leveraging mid-circuit measurements and shared unitaries to replicate the translational equivariance of classical CNNs. As an extension, Baek et al. (2022) developed a hybrid pipeline for classifying 3D point clouds. It involves voxelizing the point cloud and using quantum circuits to process dense features extracted from each voxel. Rathi et al. (2023) proposed a novel quantum autoencoder that embeds classical information through a hand-crafted approach and compresses it using partial tracing. After the compression bottleneck, the model reconstructs the information by introducing ancilla qubits initialized in their ground state.

Our QNF-Net is inspired by the two latter works. While these studies focus on classifying 3D point clouds and compression using hand-crafted embeddings only, we *shift the focus to the general field representation using learnable embeddings of classical data into quantum states, leveraging their intrinsic connection to energy representation. Additionally, we employ a carefully designed quantum circuit.* Our paper differs from many related and theoretical QML works Silver et al. (2023); Reddy & Bhattacherjee (2021); Manko & Frolovtsev (2024); Xiang et al. (2024); Blekos & Kosmopoulos (2021) in that we follow an empirical approach and scale up the supported data resolution w.r.t. them.

## 3 METHOD

This section introduces QNF-Net, our hybrid quantum framework for neural field learning; see Fig. 2 for an overview. Our goal is to learn multi-dimensional field representations such as 2D images or 3D shapes, specifically, in a coordinate-based manner. At test time, for each queried coordinate in different fields, we obtain the corresponding value encoded or inferred by the QNF-Net. We assume fault-tolerant quantum computers and focus on the fundamental challenges of the setting.

Figure 3: **Overview of our PQC architecture**: From left to right, we show: 1) parameter initializations of the quantum circuit; 2) our designed smallest repeatable PQC block structure (denoted as "Quantum Circuit"); and 3) the circuit arrangement pattern with the identity on top denoted by "$S_{R_J}$" and its adjoint "$F_{R_J}$", and Gaussian on the bottom denoted by long "$S_{G_J}$". Note that "$S$" blocks in different mentioned initialisations have the same architecture.

### 3.1 Energy Inference and Input Data Encoding

Amplitude encoding can be used to encode classical input data $x$ as a quantum state $|\psi(x)\rangle$ which can be further processed by an ansatz. One possibility of amplitude encoding is hand-crafted design, which likely can lead to sub-optimality and limitation to a single problem Rathi et al. (2023). Hence, we propose to learn the optimal density $\rho(x)_{opt}$ of quantum state encoding from data of a given type (note that we only consider pure states, i.e., $Tr(\rho(x)^2) = 1$). As quantum circuits are inherent samplers, one approach to implement a learnable amplitude encoding would be to consider a measurement outcome distribution of a quantum system and approximate it through a probabilistic neural network. However, this can easily cause challenges due to the inherent stochastic nature of the sampling process. Consequently, we adopt an alternative and new stochasticity-free approach, i.e., inferring the energy spectrum of the quantum system that automatically takes into account such probabilistic sampling uncertainty. As a first step, we employ a lightweight vanilla multi-layer perceptron (MLP) with ReLU activations to infer the energy $\boldsymbol{E}$ of the field input $\Theta$; it consists of three hidden layers with 256 neurons each. Since ReLU-based MLP is biased towards representing low-frequency signals, we incorporate positional encoding $\gamma$ to accelerate finding such energy, which is common in classical neural fields Rahaman et al. (2019); Mildenhall et al. (2021):

$$\boldsymbol{E} = \text{MLP}([\gamma(\boldsymbol{\Theta})^T, \boldsymbol{z}^T]^T), \text{ with} \tag{1}$$

$$\gamma(\boldsymbol{\Theta}) = (\sin(2^0\pi\boldsymbol{\Theta}), \cos(2^0\pi\boldsymbol{\Theta}), \cdots, \sin(2^{L-1}\pi\boldsymbol{\Theta}), \cos(2^{L-1}\pi\boldsymbol{\Theta})). \tag{2}$$

Here, $\Theta$ is our field query coordinate while $\boldsymbol{z}$ is the latent code conditioning our QNF-Net. The probability distribution $\boldsymbol{P}$ of input quantum states (originating from the encoding of classical data) can be associated with the inferred energy spectrum $\boldsymbol{E}$ using, for instance, the Gibbs-Boltzmann distribution $\varsigma$, i.e., inductive bias of our learned encoding. For a quantum system involving n qubits, and therefore, with $N=2^n$ distinct quantum states, such probability distribution $\boldsymbol{P}$ can be prepared after inferring a deterministic energy distribution $\boldsymbol{E}$:

$$P = \varsigma(\boldsymbol{E}) = \frac{e^{-\beta E(x)}}{\int e^{-\beta E(x)}dx} \approx \frac{e^{-\beta E(x)}}{\sum_{j=1}^{N} e^{-\beta E(x)_j}}, \tag{3}$$

where $\beta$ is a constant dependent on the process temperature as derived originally in thermodynamics. As $\beta$ serves as an energy scaling factor, it can be incorporated as part of the energy term and learnt without compromising the generality. As the amplitudes $\alpha_i$ of quantum states are inherently complex, their complex phases $\arg(\alpha_i)$ can take any values in the interval $[0, 2\pi)$ while satisfying the norm condition $\|\alpha_i\|_2 = \sqrt{P_i}$. Through later empirical analysis, we have observed that setting $\arg(\alpha_i) = 0$ simplifies the optimisation process while still yielding effective results. We can then prepare our final quantum encoding of the classical input data as follows:

$$|\psi_{in}\rangle = \sum_{i=1}^{N} \alpha_i |\psi_i\rangle, \quad \alpha_i = \sqrt{P_i}, \quad \hat{\rho} = |\psi_{in}\rangle\langle\psi_{in}| = \sum_{i,j=1}^{N} \alpha_i\alpha_j^+ |\psi_i\rangle\langle\psi_j|, \tag{4}$$

where $|\psi_{in}\rangle$ is our prepared quantum state , $|\psi_i\rangle$ is some local basis, and $\hat{\rho}$ describes the density distribution of $|\psi_{in}\rangle$; "$(\cdot)^+$" denotes the adjoint operator (conjugate transpose). Next, we theoretically analyse the effect of such data encoding on the expressiveness of the whole model.

**Lemma 1** *The energy inference module is functionally equivalent to finding the optimal input-weighted frequency spectrum of variational quantum circuits, which determines its expressive power.*

As proved by Schuld et al. 2021, variational circuits of the form $U(x) = W^2 g(x) W^1$ can be expressed as a truncated Fourier-type sum:

$$f(x) = \langle 0| {W^1}^\dagger g(x)^\dagger {W^2}^\dagger M W^2 g(x) W^1 |0\rangle = \sum_{w \in \Omega} c_w e^{iwx}, \tag{5}$$

where $W^1$ and $W^2$ are arbitrary unitary matrices while $g(x)$ serves as data encoding modules applied across all the qubits. Notably, $g(x)$ is general in the sense it can include data-reuploading operations without being restricted to Pauli encoding. In contrast, the encoding gates discussed in Schuld et al. 2021 specifically refer to Pauli encoding; see details in the original paper. The frequency spectrum, denoted as $\Omega$, is derived from the eigenvalues of $g(x)$. The associated Fourier coefficient $c_w$ is determined by the ansatz design parameters $W^1$ and $W^2$ and the measurement operator $M$. It is evident that our prepared input quantum state $|\psi_{in}(x)\rangle$ can be equivalently represented as $g(x)W^1 |0\rangle$ through energy inference (note: $W^1$ can be any arbitrary unitary matrix). This allows us to relate the inferred energy levels to the multi-dimensional frequency spectrum entries with the dependence encoded in the learnable weights of our energy inference framework.

### 3.2 PARAMETRISED QUANTUM CIRCUIT DESIGN

We next describe the design of our ansatz, i.e. the quantum circuit $\hat{S}(\boldsymbol{\theta})$ that induces evolution of our prepared quantum states $|\psi_{in}(x)\rangle$ as visualised in Fig. 3.1. Note that Fig. 3.1-(a) visualises the quantum circuit design included in Fig. 2. It can be noted that not all rotations around every axis in the circuit are effective as we use the expectation value of our measurements through sampling the circuit as our output; the sampled distributions depend on the evolved quantum states probabilities. For example, rotations around $Z$-axis only incur phase change, i.e., not changing probability density. Rotations around $Y$- and $X$- axis of the Bloch sphere behave similarly in modifying probability density. From this consideration and to reduce effective parameter search space, we design our circuit only out of $Y$-axis rotations, i.e., our quantum states under evolution only lie in the real-valued region as shown in Fig. 4.

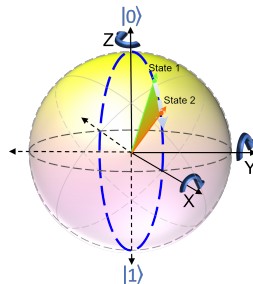

Figure 4: Possible pure qubit states and transitions between them.

We measure the qubit states after the proposed set of PQCs in the standard computational basis $\{|0\rangle, |1\rangle\}$. The measurement is performed locally, as it relieves potential barren plateau issues, i.e., $\langle \nabla_\theta L(\theta)\rangle = 0$, $\text{Var}(\nabla_\theta L(\theta)) \approx 0$, $L(\theta)$ being the loss function, for large circuits[1] Cerezo et al. (2021); Thanasilp et al. (2023). Mathematically, these local measurements translate to the following expression, for our case, involving the observable operator $\hat{O}$ chosen as the Pauli $\hat{Z}$ operator $\hat{O} = \{\hat{O}_i | \hat{O}_i = (\otimes_{k=1}^{i-1} \mathbf{1}_k) \hat{Z} (\otimes_{l=i+1}^{n} \mathbf{1}_m)\}$, where $\mathbf{1}_k$ and $\mathbf{1}_m$ denote so-called identity measurements preserving the qubit state of $i-1$ qubits before and $n-i$ qubits after the measured $i$-th qubit, respectively. Furthermore, as the observations of the quantum system inherently exhibit statistical behaviour, our final measurement output $V(x)$ is defined as the expectation value of individual qubit measurements, i.e.,

$$V(x) = Tr(\hat{\rho}(x)\hat{M}), \quad \text{and} \quad \hat{M} = \hat{S}(\boldsymbol{\theta})^\dagger \hat{O} \hat{S}(\boldsymbol{\theta}), \tag{6}$$

where $\hat{M}$ is the measurement basis. To identify the optimal measurement base $\hat{M}_{opt}$, we decompose and approximate the optimal unitary Hamiltonian evolution with single-qubit rotations and qubit entanglers. Although we know such circuit design is universal for approaching $\hat{M}_{opt}$, we still need to experiment with the required circuit layers $\mathcal{J}$, and, therefore, the total number of gates. Theorem 1 states that this required number of gates is upper bounded by $O(4^N log^4(\frac{1}{\epsilon}))$.

**Theorem 1** *Solovay-Kitaev Dawson & Nielsen (2006): Let $G$ be a finite subset of $SU(2)$ and $U \in SU(2)$. If the group is generated by $G$ is dense in $SU(2)$, then for any $\epsilon > 0$, it is possible to approximate any $U$ to precision $\epsilon$ using $O(log^4(\frac{1}{\epsilon}))$ from $G$. For multi-qubit cases, the total number of gates needed to approximate $U$ on $N$ qubits is at most $O(4^N log^4(\frac{1}{\epsilon}))$.*

---

[1]the loss function concentrates around its mean exponentially with the number of qubits

After obtaining circuit measurements $V(x)$, we still need to post-process them such that QNF-Net output $F(x)$ is consistent with the field-specific output with dimension $m$ (as we are targeting field representation problems with low output dimensions, we assume that $m$ is always smaller than the number of qubits $n$). To have this guarantee, we define a simple parity-based grouping operation $\mathcal{P}$:

$$\mathcal{P}_j = \{mk + j | k, m \in \mathbb{N}, mk + j \leq n\} \text{ and} \tag{7}$$

$$[V(x)]_i = Tr(\hat{\rho}\hat{S}(\boldsymbol{\theta})^\dagger \hat{O}_i \hat{S}(\boldsymbol{\theta})), \quad F_m(x) = \left\{ \frac{1}{|\mathcal{P}_j|} \sum_{i \in \mathcal{P}_j} [V(x)]_j \Big| j = 1, ..., m \right\}. \tag{8}$$

Note that $\mathcal{P}$ in Eq. equation 7 maps the information (statistics) from $n$ qubits to a $m$-dimensional real-valued vector, and each entry of this vector contains a possible field quantity in general.

### 3.3 QNF-NET TRAINING PARAMETER INITIALISATION

---

**Algorithm 1** QNF-Net Training Protocol

---

1: **Input:** Training data $\{x, y\}$; number of qubits $n$; epoch number $N_{\text{epochs}}$.

2: Energy inference weight initialisation: $\theta_{classical} \sim \mathcal{U}\left(-\sqrt{\frac{6}{n_{\text{in}}}}, \sqrt{\frac{6}{n_{\text{in}}}}\right)$

3: PQC initialisation: $\theta_{quantum}$ (Gaussian or identity; see Sec. 3.3)

4: **Iterative model optimisation (training) by backpropagation:**

5: **for** epoch $= 1$ to $N_{\text{epochs}}$ **do**

6:    *Classical*: Infer energy states $E$, quantum state probabilities $P$: $P_i = $ *Gibbs-Boltzmann* $(E_i)$ (see Sec. 3.1)

7:    *Quantum*: Quantum states $\hat{\rho}$ with amplitudes $\alpha_i = \sqrt{P_i}$, $arg(\alpha_i) = 0$ evolve under ansatz-induced Hamiltonian: $\hat{\rho} = |\psi_{in}\rangle \langle\psi_{in}| \rightarrow \hat{S}(\boldsymbol{\theta}) |\psi_{in}\rangle \langle\psi_{in}| \hat{S}(\boldsymbol{\theta})^\dagger$ (see Sec. 3.2)

8:    *Quantum*: Sample circuits and evaluate model outputs: $\left\{ \frac{1}{|\mathcal{P}_j|} \sum_{i \in \mathcal{P}_j} [V(x)]_j \Big| j = 1, ..., m \right\}$ (see Sec. 3.2)

9:    *Classical*: Compute loss $L(\theta)$ and gradients $\nabla_\theta L(\theta)$; backpropagate $\nabla_\theta L(\theta)$.

10: **end for**

---

Similarly to classical neural networks, where proper parameter initialisation is crucial, the quantum model requires careful parameter selection for trainability, especially for large circuits. However, initialisation protocols for QML are still developing, and only a few approaches have been proven effective; we incorporated two of them: identity and Gaussian initialisation (see Fig. 3.1-(b)).

For identity initialisation, each minimum repeatable block $\hat{M}_j$ is constructed by firstly randomly initialising trainable parameters within the interval $[0, 2\pi)$ for $S_R$ and then appending its adjoin $F_R$ such that $S_R F_R = I$ at the start of training to minimise circuit effective depth ($S_R F_R$ is not constrained to be identity in later training) Grant et al. (2019). For Gaussian initialisation, parameters $S_G$ from $\hat{M}_j$ are sampled from a zero-mean Gaussian distribution to ensure slower decay of gradient norm $||\nabla_\theta L(\theta)||_2$ with increasing circuit scale, i.e. circuit depth or number of qubits Zhang et al. (2022). Subsequently, the quantum circuit $\hat{S}(\theta)$ is built exclusively out of $\hat{R}_y$ by chaining $\mathcal{J}$ building blocks in a serial order such that the whole circuit unitary $\hat{S}(\theta)$ becomes $\hat{M}_{\mathcal{J}-1} \cdots \hat{M}_0$. The block architecture and initialisation remain consistent for all $\hat{M}_j$ within the circuit.

Consider a dataset $X$ comprising fields $X_i$ parametrised by physical field quantities $s_j$ such as pixel colours or SDF values. We prepare the field coordinates $x_j$ with $M$ sampled points as follows:

$$X_i = \{(x_j, s_j) | s_j = f(x_j), j = 0, ..., M\}. \tag{9}$$

Each $X_i$ is linked to a distinct latent code $\boldsymbol{z_i}$ initially (at the beginning of training) sampled from a zero-mean Gaussian distribution. It can be noted that maximising the likelihood $p_{\boldsymbol{\theta}}(s_j | x_j)$ is equivalent to maximising $\sum_i p_{\boldsymbol{\theta}}(s_j | x_j, \boldsymbol{z_i}) p(\boldsymbol{z_i})$. Without loss of generality, the likelihood $p_{\boldsymbol{\theta}}(s_j | x_j, \boldsymbol{z_i})$ can take the form $-\mathcal{L}(f_{\boldsymbol{\theta}}(\boldsymbol{z_i}, x_j), s_j)$, where the loss function $\mathcal{L}$ is chosen to be $\ell_1$-loss penalising disparity between predictions $f_{\boldsymbol{\theta}}(\boldsymbol{z_i}, x_j)$ and the corresponding ground-truth field values $s_j$. The prior distribution over $\boldsymbol{z_i}$ can be assumed to be a zero-mean multi-variate Gaussian function. Therefore, the loss function $L_{\boldsymbol{\theta}, \boldsymbol{z}}$ to be minimised during training with $W$ distinct fields of $M$ samples per field can be formulated as follows:

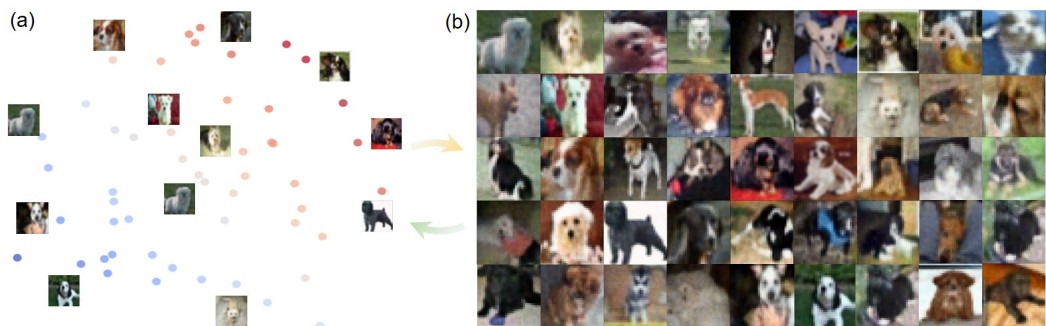

Figure 5: The latent space under 2D projection with t-SNE van der Maaten & Hinton (2008) (a) associated with reconstructed arbitrary 2D images learned by our QNF-Net (b).

$$L_{\boldsymbol{\theta},\boldsymbol{z}}(\boldsymbol{x},\boldsymbol{s}) = \sum_{i=1}^{W} \left( \sum_{j=1}^{M} \mathcal{L}(f_{\boldsymbol{\theta}}(\boldsymbol{z_i}, x_j), s_j) + \sigma^2 ||\boldsymbol{z_i}||_2 \right). \tag{10}$$

Our architecture optimisation is implemented using Adam optimiser Kingma & Ba (2015) with the initial learning rate set to $1e^{-3}$ and $N_{\text{epochs}} = 3k$ training epochs (unless mentioned otherwise). Additionally, a learning rate scheduler is employed which decays the learning rate by $10\%$ with a patience value set to 20. Our model is trained using a single A100 GPU on a high-level simulator in PyTorch provided in Pennylane Bergholm et al. (2018) with a summary of the training protocol provided in Alg. 1. We next analyse the computational complexity of simulating quantum system evolution on classical hardware. Generally, for a circuit consisting of $n$ qubits with depth $\mathcal{J}$, the non-accelerated complexity of a single coordinate query then reads $O((2^n)^3 \cdot \mathcal{J}) + O_{MLP}$; $O_{MLP}$ is the computational complexity for preparing the input quantum state through its energy state distribution. Once trained, our approach can be queried at test time by providing the query coordinates (2D or 3D) and an optional latent vector variable.

## 4 EXPERIMENTAL EVALUATION

We evaluate our approach for the representational accuracy of quantum neural fields across varying data dimensions (2D and 3D). Due to the inherent computational and memory demands associated with QML, we choose compact and representative data collections from CIFAR-10 Krizhevsky et al. (2009) and ShapeNet Chang et al. (2015) datasets. We also use some high-resolution images from James Webb Space Telescope Gardner (2022); see Fig. 1. We report the Mean Squared Error (MSE) and PSNR for 2D images and Mean Average Error (MAE) for 3D shapes.

### 4.1 COORDINATE-BASED NEURAL QUANTUM 2D IMAGE REPRESENTATION

We evaluate our QNF-Net on dense 2D image representations with pixel colours composing a 2D image field. Starting with single-image representations, we select a puppy image with rich signal frequency bands. The quantum component of our model is configured using Gaussian initialisation. To understand the learning process, we prepare the visualisation of the intermediate reconstruction results using our model along with the comparison with the MLP baseline of equivalent expressive capacity in Fig. 6 (*our MLP baselines include positional encoding and three hidden layers with 328 neurons each*). While both approaches seem to handle low-frequency information

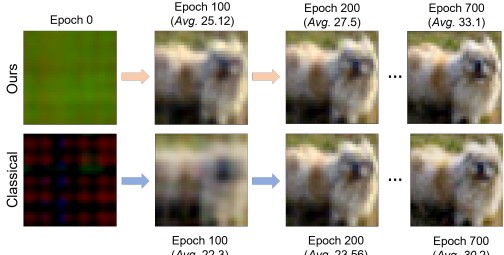

Figure 6: Reconstructed images for different epochs with average PSNR ("Avg."). QFN-Net (top) captures high-frequency details faster than the classical MLP (bottom).

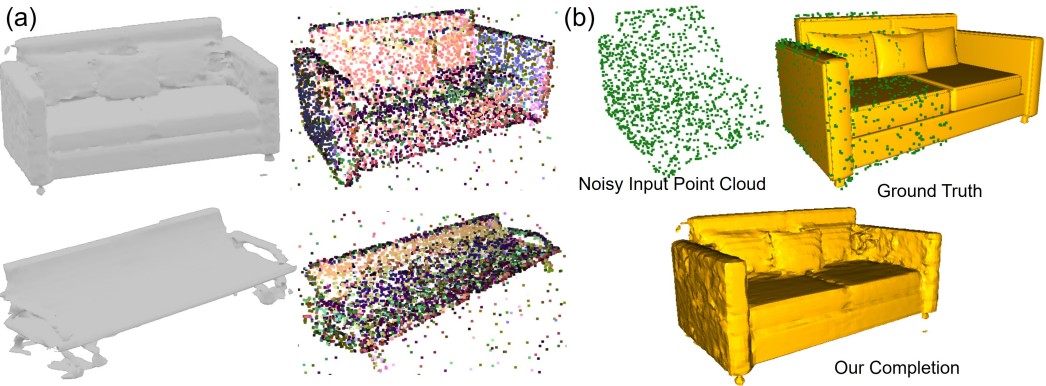

Figure 7: (a): Reconstruction of different 3D shapes using our QNF-Net. Each distinct shape is constructed using Marching Cubes based on inferred field properties from our model with their unique latent codes; samples used for reconstruction are coloured using their estimated normals. (b): Shape completion from partial inputs using QNF-Net.

well in some regions (see the grassland and the shadow on bottom-left), the MLP exhibits inferior performance for high-frequency regions (e.g. the puppy face). Our model converges faster and exhibits higher metric values such as PSNR for the same number of training epochs. We also test conditioned QNF-Net in a scenario with multiple images; see Fig. 1-(b) and Fig. 5 for examples and Table 1 for intermediate loss values across different metrics. We observe that our model significantly accelerates the learning process which is consistent with visualisations in Fig. 6.

## 4.2 COORDINATE-BASED VOLUMETRIC NEURAL QUANTUM 3D SCENE REPRESENTATION

We extend our evaluations to more challenging 3D multi-shape representations with Signed Distance Field (SDF) values with model architecture unchanged as in Sec. 4.1. By learning the 3D object surface details, i.e., signed distances, we expect our quantum neural field model to learn several shape representations simultaneously which can be used for other downstream tasks. We initially sample $200k$ points with SDF values for our 3D field with higher sampling density near the surface for a higher level of object detail capture. However, we encounter memory depletion issues on our hardware due to storage requirements of intermediate results and gradient-related numerics upon our quantum component, which is presumably due to uncomparable optimisation and maturity level compared with current deep learning tools. We, therefore, experiment using six 3D shapes with each represented by $100k$ sample points at the expense of reduced reconstruction quality; some visual results extracted using Marching Cubes Lorensen & Cline (1998) with inferred field values are provided in Fig. 7-(a).

Table 1: Numerical comparison of reconstruction performance between our approach and a standard MLP baseline. "co." means converged. "clas/quant" means classical and quantum.

| Method | Epoch | # Params (clas/quant) | Images (MSE↓/ PSNR↑) | | 3D Shapes (MAE↓) | |
|---|---|---|---|---|---|---|
| | | | w/ PE L = 6 | w/o PE | w/ PE L = 6 | w/o PE |
| Ours [Gaussian] | 100 200 3k (co.) | 1.56e5/ 120 | 1.6 e-2/ 17.74 1 e-2 / 19.96 **1 e-3 / 29.2** | 3.4 e-2/ 14.64 2.8 e-2/ 15.52 **3 e-3 / 25.37** | 1.9 e-3 1.6 e-3 **1 e-3** | 2.3 e-3 1.9 e-3 1.6 e-3 |
| Ours [Identity] | 100 200 3k (co.) | 1.56e5/ 120 | 1.9 e-2 / 17.23 9 e-3/ 20.36 1 e-3 / 28.8 | 3.4 e-2/ 14.63 2.4 e-2 / 16.19 3 e-3 / 24.68 | 1.8 e-3 1.6 e-3 1.1 e-3 | 2.3 e-3 2 e-3 **1.5 e-3** |
| MLP Baseline | 100 200 3k (co.) | 2.2e5 | 4.4 e-3/ 13.5 1.8 e-2/ 17.42 2 e-3/ 26.57 | 6.7 e-2 / 11.72 3.5 e-2/ 14.51 1.1 e-2/ 19.58 | 2.7 e-3 1.9 e-3 1.4 e-3 | 3.5 e-3 2.6 e-3 2e-3 |

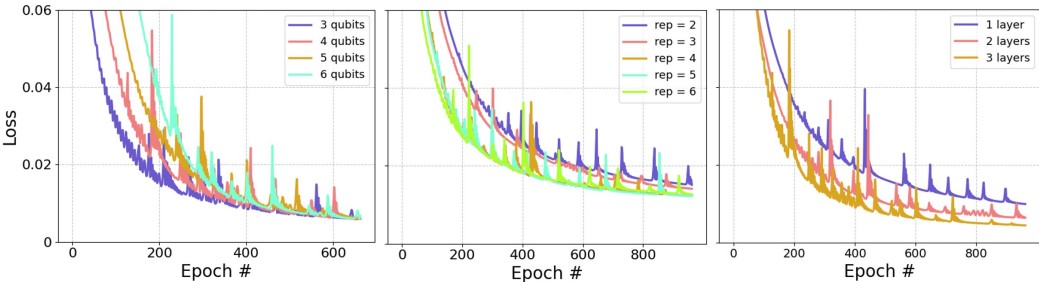

Figure 8: Ablation experiments on modules that influence the training performance. Here, our experiments include the number of involved qubits $n$ (left), building block repetitions $\mathcal{J}$ (middle) and classical energy inference layers (right). Note that the model with more qubits trains slower but supports a higher resolution of the learned signals (2D-pixel grids and sampled surfaces for SDFs).

### 4.2.1 Image Inpainting and Partial Shape Completion

A key advantage of our quantum neural field representation is its ability to perform inference with partial samples. This enables partial shape completion or image inpainting by identifying a latent code $\hat{z}$ that best explains the partial observation $X_i$ while keeping the pre-trained model parameters fixed, using Maximum-a-posteriori (MAP) estimation as follows:

$$\hat{z} = \underset{z}{\arg\min} \sum_{(x_j, s_j) \in X_i} \mathcal{L}(f_{\boldsymbol{\theta}}(\boldsymbol{z}, x_j), s_j) + \sigma^2 ||\boldsymbol{z}||_2. \tag{11}$$

Once $\hat{z}$ is determined, we can sample from QNF-Net in the entire space conditioned on $\hat{z}$ to obtain the complete 3D shape or 2D image as shown in Fig. 7-(b) and Supplement B.

### 4.3 Ablation Study

Besides, we also perform ablation studies analysing the effects of different sub-modules, e.g. the number of involved qubits $n$, circuit depth $J$ and the expressivity of energy inference module (see Fig. 8). With increasing circuit depth $J$, ranging from 2 in ascending order to 6 (middle figure), the learning becomes more efficient and accurate in terms of converged value, i.e., consistent with Solovay-Kitaev theorem. Additionally, augmenting the expressivity of energy inference modules through the inclusion of more hidden layers leads to both improved convergence and better performance (right figure). However, as the number of used qubits $n$ increases, this improvement gets countered in a noticeable way (left figure). We suspect that this can stem from the increased complexity of identifying the optimal energy description of the problem input while other factors are strictly controlled. This implies that we do not need to manipulate many qubits, which can be advantageous in practice and on upcoming quantum processors.

### 4.4 Adaption to other energy inference network designs

We want to highlight the flexibility in designing the energy inference module. For instance, using a sinusoidal activation function instead of the conventional ReLU can improve convergence to the optimal energy setup. This approach is empirically evaluated alongside classical SIREN Sitzmann et al. (2020), i.e., vanilla-type dense multi-layer

Table 2: Comparisons against SIRENs.

| Method | Epoch | Images (MSE)↓ | 3D Shapes (MAE)↓ |
|---|---|---|---|
| Ours (periodic activation) | 200 | 3.4 e-3 | 8 e-4 |
| | co. | **7.8 e-4** | **2.7 e-4** |
| SIREN | 200 | 8.1 e-3 | 1.3 e-3 |
| | co. | 1 e-3 | 4.8 e-4 |

network with a periodic activation function. While this work primarily focuses on the whole framework design, other viable design approaches that could infer the energy more efficiently depending on specific scenarios could also be integrated. The results are summarised in Tab. 2.

## 5 DISCUSSION AND CONCLUSION

This paper formulates a new quantum framework for encoding classical data adaptively into quantum states, following entirely new principles compared to existing literature, i.e., *quantum system evolution and measurements on quantum devices*, which can be used for tasks related to neural field representations. We observe in different scenarios that the QNF-Net allows us to improve both the convergence speed and the representation accuracy compared to different baselines; it can even challenge stronger baselines by incorporating more advanced energy-inference network designs (SIREN). At the same time, it also supports several useful real-world tasks such as shape completion and interpolation in the latent space. We theoretically analyse the model and perform experiments on large-scale 2D and 3D datasets *while previous QML works evaluate on small-resolution images*. Moreover, we perform thorough ablation studies of different module components. Notably, we do not observe barren plateaus, thanks to our design choices. The contributing factors to their absence can be manifold such as measurement locality, network hybridisation, and weight initialisation.

While our work highlights the potential of QML in general neural field representation, there are avenues for future research. Our current approach only partially leverages the information from the Hilbert space due to optimisation and circuit complexity considerations; see Fig. 4. Developing more effective encoding strategies or deploying scenario-conditioned network model design to harness additional Hilbert space information while not compromising optimisation performance would be a promising direction for further study. While a standard, widely accepted approach for efficient amplitude encoding of arbitrary classical normalized data has yet to emerge, the field of its physical realizations is advancing at a remarkable pace Ashhab (2022); Gonzalez-Conde et al. (2024); Daimon & Matsushita (2024). The proposed innovative association between amplitude encoding and energy could potentially inspire the preparation of such quantum states with Hamiltonian evolution using devices such as quantum annealers. Furthermore, future research could explore alternative data encoding strategies that could become more practical as the implementation of QNF-Net on real quantum hardware becomes feasible. Even in the absence of suitable quantum hardware in the near term, our QNF remains valid as a quantum-inspired method. While it does not yet account for all aspects of hardware realization, it offers valuable insights and progresses the field. Lastly, beyond the demonstrated applications in shape interpolation and completion, QNF also shows promise for tasks such as image and shape classification.

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
