# Appendix

This appendix provides comprehensive background knowledge on gate-based quantum computing and its derivations from quantum physics (see Sec. A).

## A  BACKGROUND

### A.1  PRELIMINARIES OF GATE-BASED QUANTUM COMPUTING

**Qubits**. The fundamental information blocks of a quantum processing unit (QPU) are qubits, i.e., the analogues of bits in classical computing. Unlike classical bits deterministically representing one possible state (0 or 1), qubits can statistically represent two distinct information states at the same time, denoted in the bra–ket notation as $|0\rangle$ and $|1\rangle$.

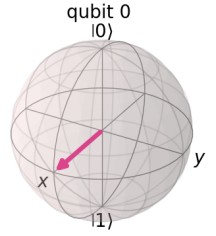 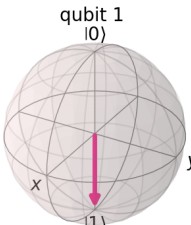

Figure 9: Bloch sphere visualisation of qubit states. Qubit 0: $|\psi\rangle = \frac{1}{\sqrt{2}}(|0\rangle + |1\rangle)$, qubit 1: $|\psi\rangle = |1\rangle$.

*Superposition* is a fundamental property distinguishing qubits from bits: It grants qubits the capacity to exist in a combinatorial state $|\psi\rangle$ of $|0\rangle$ and $|1\rangle$ such that:

$$|\psi\rangle = \alpha |0\rangle + \beta |1\rangle, \tag{12}$$

with $\alpha, \beta \in \mathbb{C}$ and $|\alpha|^2 + |\beta|^2 = 1$. Qubit states $|\psi\rangle$ can be visualised on Bloch spheres (see Fig. 9) or expressed in vector forms:

$$|0\rangle = \begin{bmatrix} 1 \\ 0 \end{bmatrix}, |1\rangle = \begin{bmatrix} 0 \\ 1 \end{bmatrix}, |\psi\rangle = \alpha |0\rangle + \beta |1\rangle = \begin{bmatrix} \alpha \\ \beta \end{bmatrix}. \tag{13}$$

*Measurement* in quantum mechanics inherently adopts a statistical approach to extract numerical information. For a qubit state $|\psi\rangle = \alpha |0\rangle + \beta |1\rangle$ measured with operator $\hat{O}$ (that must be Hermitian, i.e., $\hat{O}^\dagger = \hat{O}$), this implies probabilities $|\alpha|^2$ and $|\beta|^2$, respectively, for measuring the information (i.e., eigenvalue of the measurement operator $\hat{O}$) stored in states $|0\rangle$ and $|1\rangle$:

$$\hat{O} |0\rangle = \kappa |0\rangle \text{ and } \hat{O} |1\rangle = \delta |1\rangle. \tag{14}$$

$\kappa$ and $\delta$ are the statistical information that can be measured, i.e., eigenvalue of the measurement operator $|O\rangle$. A key aspect of measurement is the phenomenon known as wave function collapse, i.e., the projective measurement causes $|\psi\rangle$ to collapse to the operator's eigenstate, $|0\rangle$ or $|1\rangle$, conditioned on the measurement, i.e., $\kappa$ or $\delta$.

*Entanglement* is considered to be another potential advantage of quantum computing over classical computing. In classical computing, information stored in bits is independent, i.e., measuring one bit does not affect others. In the quantum realm, qubits can be highly correlated, exhibiting entanglement such that the information of one qubit can be interrelated with another despite distance. For instance, a general information state of a 2-qubit system $|\psi\rangle_2$ can be expressed as:

$$|\psi\rangle_2 = a |00\rangle + b |01\rangle + c |10\rangle + d |11\rangle. \tag{15}$$

With $a, b, c, d \in \mathbb{C}$ such that $|a|^2 + |b|^2 + |c|^2 + |d|^2 = 1$. The 2-qubit system is considered entangled if $|\psi\rangle_2$ cannot be expressed as a tensor product of two qubits $|\psi\rangle_{a1}$ and $|\psi\rangle_{a2}$, indicating that their information cannot be independently measured without disturbing each other, i.e.,

$$|\psi\rangle_2 \neq |\psi\rangle_{a1} \otimes |\psi\rangle_{a2}. \tag{16}$$

**Rotation Operators**. The operators responsible for rotating quantum states $|\psi\rangle$ of qubits along $x, y, z$ axes on a Bloch sphere are referred to as rotation operators. Any single qubit operator $\hat{R}$ can be expressed as a combination of such rotation operators $\hat{R}_x, \hat{R}_y, \hat{R}_z$, i.e., $\hat{R}(\theta, \tau, \gamma) = \hat{R}_x(\theta)\hat{R}_y(\tau)\hat{R}_z(\gamma)$ with angles $\theta, \tau$ and $\gamma$:

$$\hat{R}_x(\theta) = \begin{bmatrix} cos(\frac{\theta}{2}) & -isin(\frac{\theta}{2}) \\ -isin(\frac{\theta}{2}) & cos(\frac{\theta}{2}) \end{bmatrix}, \tag{17}$$

$$\hat{R}_y(\tau) = \begin{bmatrix} cos(\frac{\tau}{2}) & -sin(\frac{\tau}{2}) \\ sin(\frac{\tau}{2}) & cos(\frac{\tau}{2}) \end{bmatrix}, \tag{18}$$

$$\hat{R}_z(\gamma) = \begin{bmatrix} e^{-i\frac{\gamma}{2}} & 0 \\ 0 & e^{i\frac{\gamma}{2}} \end{bmatrix}. \tag{19}$$

The Pauli operators $\hat{X}, \hat{Y}, \hat{Z}$ represent specific instances of rotation operators, inducing rotations by $\pi$ radians along the $x, y, z$ axes, respectively. These operators can also be expressed as matrices in the computational basis $|0\rangle, |1\rangle$ as follows:

$$\hat{X} = \begin{bmatrix} 0 & 1 \\ 1 & 0 \end{bmatrix}, \hat{Y} = \begin{bmatrix} 0 & -i \\ i & 0 \end{bmatrix}, \hat{Z} = \begin{bmatrix} 1 & 0 \\ 0 & -1 \end{bmatrix}. \tag{20}$$

**Schrödinger's Equation**. Quantum computing involves the manipulation of information according to the principles of quantum mechanics, with its foundation rooted in time-dependent Schrödinger's equation:

$$i\hbar\frac{d}{dt}|\psi(t)\rangle = \hat{H}(t)|\psi(t)\rangle, \tag{21}$$

where $\hbar$ is Planck's constant while $|\psi(t)\rangle$ and $|\psi(0)\rangle$ are the quantum states after and before evolution, respectively. $\hat{H}$ is the Hamiltonian operator of the quantum system. Therefore, the evolution of quantum states can be described by the following relationship:

$$|\psi(t)\rangle = \hat{T}e^{-\frac{i}{\hbar}\int_0^t \hat{H}(t)dt}|\psi(0)\rangle, \tag{22}$$

with $\hat{T}$ denoting the time ordering operator. This simplifies to $e^{-\frac{it}{\hbar}\hat{H}}|\psi(0)\rangle$ for time-independent $\hat{H}$. Using a more compact notation, Schrödinger's Equation can also be equivalently written as:

$$|\psi(t)\rangle = \hat{U}(\hat{H}, t)|\psi(0)\rangle, \text{ with} \tag{23}$$

$$\hat{U}(\hat{H}, t) = e^{-\frac{it}{\hbar}\hat{H}}. \tag{24}$$

To perform rotation operations on qubits, the system Hamiltonian $\hat{H}$ can be set to $E\hat{\sigma}$ with $\hat{\sigma} \in \{\hat{X}, \hat{Y}, \hat{Z}\}$ and by setting $\eta = 2Et/\hbar$, we have:

$$\hat{U}(\hat{H}, t) = e^{-\frac{it}{\hbar}\hat{H}} = e^{-\frac{i\alpha}{2}\hat{\sigma}} = \hat{R}_\sigma(\eta). \tag{25}$$

## A.2 BARREN PLEATAU

When training a quantum ansatz $\hat{S}(\boldsymbol{\theta})$, employing a native unbiased random initialisation could potentially lead to training issues due to the concentration of measure, as argued by McClean et al. 2018:

*"...for a wide class of reasonable parameterised quantum circuits, the probability that the gradient along any reasonable direction is non-zero to some fixed precision is exponentially small as a function of the number of qubits."*

This observation is also known as "*barren plateau*" which can be expressed mathematically for a system with $n$ qubits as:

$$\mathbb{E}_w[\partial_w L(w)] = 0, \tag{26}$$

$$\text{Var}_w[\partial_w L(w)] \in O(\frac{1}{\nu^n}), \ \nu > 1. \tag{27}$$

This poses challenges, particularly for gradient-based learning strategies. Identified factors contributing to the barren plateau phenomenon include the locality of observables Cerezo et al. (2021); Thanasilp et al. (2023), specific noise models Wang et al. (2021), and ansatz close to a 2-design, i.e., matching Haar random unitaries up to the second moment McClean et al. (2018); Holmes et al. (2022). This highlights the importance of selecting appropriate initialisation protocols, quantum ansatz designs, and observables.

To relieve the trainability problem, Grant et al. 2019 proposed to boost the gradient flow through reducing effective circuit depth initially by employing identity blocks. Zhang et al. 2022 suggested that with proper Gaussian initialisation, the gradient norm decays at most polynomially as a function of qubit count and circuit depth applicable to both local and global observables. Cerezo et al. 2021 theoretically analysed this phenomenon from the locality view of information extraction and demonstrated barren plateaus could be avoided by using cost functions that only have information extracted from part of the circuit.

### A.3 QUANTUM MACHINE LEARNING (QML)

The expectations that quantum computing can enhance machine learning algorithms have given rise to Quantum Machine Learning (QML) Schuld et al. (2015). QML integrates principles of quantum mechanics into machine learning to tackle computationally intensive or inherently challenging problems. By replacing classical artificial network modules with parametrised quantum circuits, QML aims to enhance information processing efficiency and alleviate biases inherent in classical models Biamonte et al. (2017). A QML approach includes two stages, i.e. input encoding or a *feature map* and parametrised quantum circuit or an *ansatz*. To learn optimal parameters in the circuit, hardware-dependent gradient-based or gradient-free strategies can be used Bergholm et al. (2018); Mitarai et al. (2018); Guerreschi & Smelyanskiy (2017). Similar to classical neural networks, quantum networks are shown to be universal function approximators Benedetti et al. (2019); Schuld et al. (2021). Various quantum analogues of machine learning algorithms have been explored, including quantum principle component analysis Lloyd et al. (2014), quantum support vector machine Rebentrost et al. (2014), quantum Boltzmann machine Amin et al. (2018) and quantum k-means Kerenidis et al. (2019). Quantum gates are applied on an input quantum state generated by a feature map Kwak et al. (2021).

**Input Encoding (Feature Map)**. Classical data $x$ must be encoded into quantum states $|\psi(x)\rangle$ through a feature map to enable processing within the quantum circuit. Various established methods exist for achieving this, including basis encoding, time-evolution encoding, amplitude encoding, Hamiltonian encoding, and others. However, the question of optimal encoding across different problems is still open.

**Parametrised Quantum Gates (Ansatz)**. Quantum states $|\psi(x)\rangle$ with embedded classical information need to be processed by unitary quantum gates $\hat{U} \in \mathbb{C}^{2^n \times 2^n}$ parametrised by $\theta$ acting on $n$ qubits. In real devices, a quantum circuit (or ansatz) is composed of such unitary operations in a certain order $\hat{U}(\boldsymbol{\theta}) = \mathcal{T}(\prod_{i=1}^{t} \hat{U}_i(\theta))$; $\mathcal{T}$ is the ordering operator. A quantum circuit $\hat{U}(\boldsymbol{\theta})$ maps a quantum state $|\psi(x)\rangle$ to a new state $|\phi(x)\rangle$, i.e., $|\phi(x)\rangle = \hat{U}(\boldsymbol{\theta}) |\psi(x)\rangle$.

**Measurement**. With the evolution of quantum states under the ansatz, classical data can be extracted statistically from a quantum state $|\phi(x)\rangle$ using Hermitian measurement operators $\hat{O}$. Due to the inherent statistic process, the output $V(x)$ is normally defined as the expectation value of statistical measurements: $V(x) = \langle\phi(x)| \hat{O} |\phi(x)\rangle$. Typically, this step occurs at the end, as it collapses the embedded information conditioned on the measured value; there are rare exceptions for special purposes Gili et al. (2023); Cong et al. (2019).

**Quantum Model Training**. In line with all learning-based methodologies, QML follows a similar paradigm. It involves adjusting the parameters of the quantum ansatz either on quantum hardware or classical simulators to minimise a predefined cost function $L(\theta)$ using optimisation techniques; see supplement A.2 for details. However, efficient training in this domain remains a nascent field, undergoing active research and development.

An alternative option is the finite difference method, which provides an approximate rather than analytical gradient. However, limited by the higher computational expense associated with gradi-

ent calculation on real quantum devices, simulating such a process is often preferable for exploring medium-to-large scale problems. When using simulators, quantum information can be extracted in a single shot, making it plausible to use conventional gradient calculation methods like backpropagation.

## B  IMAGE INPAINTING RESULTS

Similar to 3D shape completion experiment results as shown in Fig. 7-b, such experiments can also be performed for 2D image inpainting. We provide noisy images and formulate the search for a latent vector that best explains the provided noisy field values as an optimisation problem, and restore the complete noiseless field. Partial results are shown in Fig. 10.

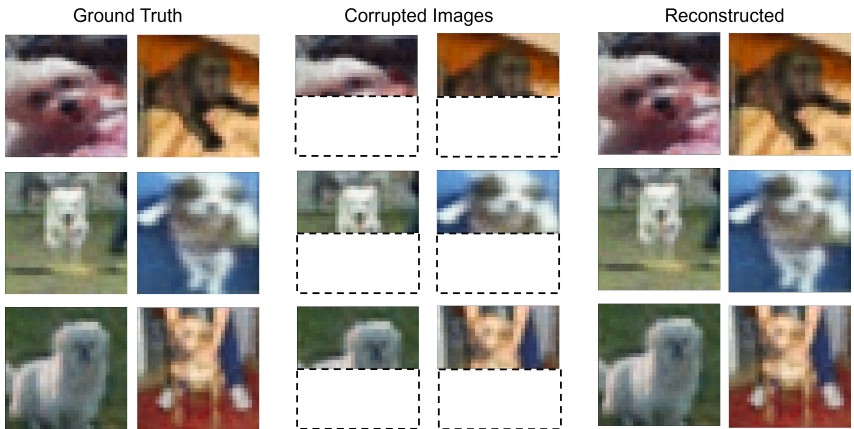

Figure 10: Image inpainting results from partial field values using QNF-Net.

## C  SHAPE COMPLETION FROM PARTIAL AND NOISY INPUT DEPTH MAPS

To further evaluate our method, we introduce zero-mean Gaussian noise to the clean depth maps and assess its impact on shape completion performance. We vary the perturbation ratio denoted as $\alpha$, across the following values: 0, 0.005, 0.01, 0.02, and 0.03. The corresponding completion results are illustrated in Fig. 11. As evident from the figure, the quality of shape completion progressively deteriorates with the increasing noise levels.

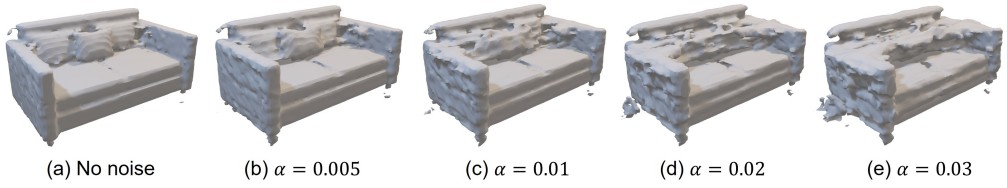

(a) No noise     (b) $\alpha = 0.005$     (c) $\alpha = 0.01$     (d) $\alpha = 0.02$     (e) $\alpha = 0.03$

Figure 11: Shape completion results from partial and noisy input depth maps using QNF-Net.

## D  MORE RESULTS

In this section, we present further empirical results to complement those provided in the main text. From the outset, we designed our experimental framework to ensure scalability, allowing us to handle large datasets, including 3D shapes, despite the current limitations of reliable quantum hardware and efficient software simulators. While our QNF-Net model enables us to scale up experiments to 3D datasets, however, the number of available 3D shapes remains constrained, depending on the dataset size, i.e., sampling density and number of shapes, as explained in the main text (see Sec. 4.2). Since the results in the main text reflect the average performance across these experiments, we also

Table 3: Detailed numerical reconstruction results for more 2D images.

| Image Batch | Epoch | Ours (Gaussian) (MSE ↓ / PSNR ↑) | Ours (Identity) (MSE ↓/ PSNR ↑) | MLP Baseline (MSE ↓/ PSNR ↑) |
|---|---|---|---|---|
| Batch 1 | 100 | 0.0183 / 17.37 | 0.0164 / 17.85 | 0.042 / 13.76 |
| | 200 | 0.012 / 19.21 | 0.0138 / 18.6 | 0.0198 / 17.03 |
| | co. | 0.001 / 30.2 | 0.001 / 29.8 | 0.0017 / 27.69 |
| Batch 2 | 100 | 0.0165 / 17.82 | 0.021 / 16.77 | 0.048 / 13.18 |
| | 200 | 0.01 / 19.3 | 0.015 / 18.24 | 0.017 / 17.69 |
| | co. | 0.0012 / 29.21 | 0.0012 / 29.07 | 0.002 / 26.9 |
| Batch 3 | 100 | 0.0159 / 17.98 | 0.023 / 16.38 | 0.035 / 14.56 |
| | 200 | 0.013 / 18.86 | 0.011 / 19.58 | 0.0186 / 17.3 |
| | co. | 0.0017 / 27.69 | 0.0015 / 28.24 | 0.0022 / 26.57 |
| Batch 4 | 100 | 0.0192 / 17.16 | 0.017 / 17.69 | 0.04 / 13.97 |
| | 200 | 0.0176 / 17.54 | 0.013 / 18.86 | 0.017 / 17.69 |
| | co. | 0.0014 / 28.53 | 0.001 / 30.44 | 0.0016 / 28.95 |
| Batch 5 | 100 | 0.0223 / 16.51 | 0.02 / 16.9 | 0.037 / 14.32 |
| | 200 | 0.007 / 21.55 | 0.018 / 17.44 | 0.0176 / 17.54 |
| | co. | 0.0008 / 30.97 | 0.001 / 29.5 | 0.0019 / 27.21 |
| *Avg.* performance | 100 | 0.0184 / 17.37 | 0.0195 / 17.12 | 0.04 / 13.96 |
| | 200 | 0.012 / 19.29 | 0.014 / 18.54 | 0.018 / 17.45 |
| | co. | 0.0012 / 29.32 | 0.001 / 29.41 | 0.0019 / 27.46 |

Table 4: Detailed numerical reconstruction results for more 3D shapes.

| 3D Shapes Batch | Epoch | Ours (Gaussian) (MAE ↓) | Ours (Identity) (MAE ↓) | MLP Baseline (MAE ↓) |
|---|---|---|---|---|
| Batch 1 | 100 | 0.0018 | 0.002 | 0.0024 |
| | 200 | 0.00156 | 0.0015 | 0.0018 |
| | co. | 0.0009 | 0.0012 | 0.0015 |
| Batch 2 | 100 | 0.002 | 0.0018 | 0.0026 |
| | 200 | 0.0016 | 0.00143 | 0.00163 |
| | co. | 0.001 | 0.0011 | 0.0011 |
| Batch 3 | 100 | 0.0016 | 0.0021 | 0.0026 |
| | 200 | 0.0015 | 0.0016 | 0.00194 |
| | co. | 0.001 | 0.001 | 0.0014 |
| Batch 4 | 100 | 0.002 | 0.0019 | 0.00276 |
| | 200 | 0.0016 | 0.0017 | 0.002 |
| | co. | 0.0008 | 0.0009 | 0.00135 |
| Batch 5 | 100 | 0.0017 | 0.0018 | 0.0025 |
| | 200 | 0.00153 | 0.00167 | 0.0019 |
| | co. | 0.0009 | 0.0012 | 0.00156 |
| *Avg.* performance | 100 | 0.00182 | 0.00192 | 0.00257 |
| | 200 | 0.00156 | 0.00158 | 0.0018 |
| | co. | 0.0009 | 0.0011 | 0.0014 |

include numerical details for all training images and shape batches in Tables 3 and Tab. 4. They provide further insights into the model's behaviour.

# E    ADDITIONAL 3D SHAPE VISUALISATIONS

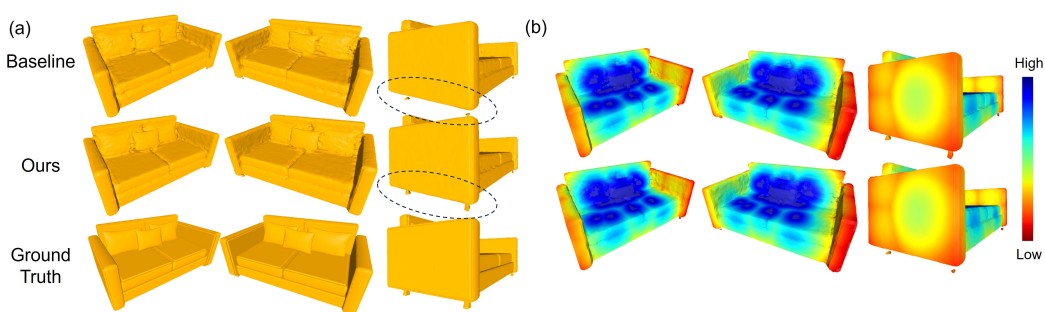

Figure 12: (a): Single shape, i.e., a sofa, reconstructed by our approach and the classical baseline from different views. The ground truth is presented at the bottom; (b) comparison of classically reconstructed 3D shape (bottom) against our approach (top) visualized via Hausdorff distance from various perspectives. The rendered image employs a colour gradient (blue>green>yellow>red) denoting descending Hausdorff distance levels.

# F    VISUALIZATION OF WEIGHTS EVOLUTION FOR THE QUANTUM MODULES

To quantify the changes in the tunable parameters of the quantum layer during training and evaluate their significance within the overall model pipeline, we present a detailed visualization of the weight distributions in the quantum layer before and after training, as shown in Fig. 13, with Gaussian initialization as the starting point. This analysis offers insights into the role of the quantum component and the dynamics of training, complementing our previous experimental findings. During training, we

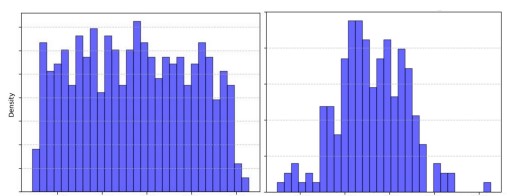

Figure 13: Histogram showing the tunable parameters of the quantum layer after (left) and before (right) training.

observed that the initialized tunable parameters in the quantum layer begin to de-concentrate around the Gaussian mean while maintaining relatively stable distribution bounds.