# OpenReview forum: "Quantum Neural Fields"
_ICLR.cc/2025/Conference — Submitted to ICLR 2025_

### Official Review · Reviewer_McH9 · 2024-10-28

**Soundness:** 3
**Presentation:** 1
**Contribution:** 3
**Rating:** 6
**Confidence:** 3

**Summary:**

The paper introduces QNF-Net, a quantum neural field network for efficiently encoding and representing 2D images and 3D shapes. Using a unique neuro-deterministic encoding method, QNF-Net maps classical data into quantum states, achieving faster convergence and higher accuracy than classical models. It performs well on visual tasks like image rendering and 3D shape reconstruction, highlighting potential quantum advantages in efficiency and scalability for large-scale visual data.

**Strengths:**

1. The paper presents a novel alternative to the classical neural field with the help of quantum computing.
2. The paper empirically demonstrates that the method is superior to the classical counterpart, making it a promising direction to the field.

**Weaknesses:**

1. Consider the novelty of this method. Are there any considerations when applying this method to the current NISQ devices? No related experiments are shown in this work.
2. This paper does not show time—and space-related complexity, which further concerns the practicality of this algorithm on classical simulators and real quantum devices.

**Questions:**

1. What is the computational complexity of this method?
2. What are the computational resources required for this method?
3. What are the impacts of this method on the effect of noise in the quantum devices?
4. I haven't seen any experiments on real quantum devices; are there any relevant experiments?

---

> ### Author Response · Authors · 2024-11-23
>
> Thank you for your insightful and valuable comments! We are delighted by your recognition of the technical novelty in combining quantum machine learning (QML) with neural implicit representations. We appreciate your acknowledgement of its significance to the rapidly advancing field of implicit neural representations and its potential to drive progress, particularly in visual computing. Below, we carefully address your questions and concerns.
>
> **question 1:**
>
> The computational complexity of this method running on a classical simulator is detailed between L320 and L323.  When running a quantum circuit on physical hardware, analyzing its complexity is challenging as it depends on factors such as gate fidelity, hardware architecture, gate types and physical implementation etc. The fidelity and noise levels impact accuracy and may require additional corrections, while hardware-specific constraints like qubit connectivity can increase circuit depth. Different quantum systems (e.g., superconducting, trapped ions) impose varying demands on gate execution speed and reliability, and limited coherence times restrict the number of gates that can be executed. These factors make it hard to analyze the complexity of running quantum circuits on physical hardware.
>
> **question 2:**
>
> The computational resources required by this method depend on factors such as sampling density, model size, and others. In our setup, training the model on 2D images and 3D scenes takes approximately 3 hours and 8 hours, respectively, on an A100 GPU.
>
> **question 3:**
>
> The primary goal of this work is to explore an efficient model that can be applied to large-scale datasets while leveraging the advantages of our quantum circuit and processing, assuming we have a fault-tolerant quantum hardware. Therefore, we assumed no noise in this study. However, with presence of noise, we would anticipate a significant performance degradation, as demonstrated by other works which primarily investigate the influence of noise on machine learning; see Borras et al. Journal of Physics: Conference Series.Vol.2438.No.1.IOP Publishing, 2023.
>
> **question 4:**
>
> Given our focus on exploring the potential of these models and their applications to large-scale datasets, particularly dense 3D reconstructions, we have chosen to rely on high-end simulators rather than NISQ hardware. While there have been notable advancements in circuit-based quantum hardware, the availability of devices with a sufficient number of stable, high-quality qubits remains extremely limited. Besides, our attempt to use IBM’s circuit-based quantum computer was hindered by unrealistic queuing time. Despite the clear importance of testing on real quantum hardware, current limitations make classical simulations the most viable option to overcome the constraints of NISQ devices, even for recent research involving smaller datasets; see Rathi et al. "3D-QAE: Fully Quantum Auto-Encoding of 3D Point Clouds." (2023); Baek et al. "3D scalable quantum convolutional neural networks for point cloud data processing in classification applications." arXiv preprint arXiv:2210.09728 (2022).

---

> > ### Comment · Reviewer_McH9 · 2024-11-27
> >
> > I understand these responses have resolved my concerns.

---

### Official Review · Reviewer_DCFN · 2024-10-31

**Soundness:** 1
**Presentation:** 1
**Contribution:** 1
**Rating:** 1
**Confidence:** 5

**Summary:**

This work presents a quantum version of the neural field named "QNF-Net". QNF-Net is claimed to have potential quantum advantages and can be applied to many scenarios, including robotics and 3D reconstruction. They introduce a method to translate classical data into quantum states, followed by a parameterized quantum circuit to train the neural field network.

**Strengths:**

Neural field network is a good model in CG/CV with growing interest. So it will be interesting to implement a quantum-enhanced neural field network.

**Weaknesses:**

1. Confusing logic.

To be honest, I do not think the logic of this paper is clear. For example, from line 083 to line 086, the discussion and remarks on the previous works, the challenges for quantum machine learning, and the methods of this paper ALL exist in THE SAME sentence, making it hard to figure out what are the authors' contributions.

2. Insufficient literature review.

In section 2 "Related work", the authors mention many other quantum machine learning algorithms for CV/CG. However, despite feeble criticism, I am afraid that I can not get any insight into other works. In particular, I can not find the difference between this paper and previous works. I think the authors should spend more time to explain also other works and the contribution of this paper.

3. Unclear and undefined algorithm procedures.

There are many black-box algorithm subroutines that are unclear or undefined: For example, the so-called "inferred energy spectrum E" is claimed to play a central role in data encoding. However, the MLP to infer such energy is unclear: what is the MLP structure, what is the loss function, and what is the training procedure?

4. Unfriendly paper writing.

(1) Inconsistent terminologies and undefined notations.

There are many inconsistent and informal notations so it is sometimes confusing for the reader to follow. For example, there are 73 "neural" and 3 "neuro", as well as 41 "circuit" and 4 "circuitry". I do not think that there is an explanation for the reader to distinguish between those confusing terminologies. Further, undefined notations can be found everywhere in this paper.

(2) Poor grammar and long sentences with no comma.

There are many grammar and punctuation errors, making this paper hard to follow, e.g., "training neural fields can be computationally and resource-demanding" in line 045,  "All these applications became possible in recent years, as there has been a notable shift from hand-crafted priors, primarily based on heuristics, to learning priors in the form of neural fields directly from data Xie et al. (2022b), with multi-layer perceptron (MLP) with ReLU activation being one popular building block for such a neural field, in the early days." from line 040 to line 043, "as most other work do" in line 083.

**Questions:**

I do not have any further questions.

---

> ### Author Response · Authors · 2024-11-23
>
> Thank you for your time and for providing valuable comments. We are encouraged by your acknowledgement of our work as a good model in computer vision and graphics and its compellingness. As we did not find any concrete questions in the review, we carefully address the high-level concerns next.
>
> **Concern 1: confusing logic**
>
> Thank you for bringing this to our attention. We are somewhat surprised, as this concern has not been raised by other reviewers. Our contributions are clearly outlined in lines 89 to 99.  We kindly encourage you to review this section thoroughly.
>
> **Concern 2: Insufficient literature review**
>
> We have enhanced the related work section by thoroughly discussing and analysing each referenced study. The distinctions between our work and existing research are now more clearly articulated.
>
> **Concern 3: unclear and undefined algorithm procedures**
>
> The algorithmic protocol (from L287 to L294) serves as a high-level guide for readers. Specifically, we use basic vanilla fully-connected multi-layer perceptrons to test the effectiveness of our hybrid model. The training procedure and loss function are clearly defined in Section 3.3. We kindly encourage you to review this section thoroughly.
>
> **Concern 4: Unfriendly paper writing**
>
> We have addressed the inconsistencies you mentioned and replaced "circuitry" with "circuit." Regarding the use of "neuro" and "neural," we kindly ask you to consider the different contexts in which each term should be used. Specifically, "neural" is an adjective, while "neuro" is a prefix.
> Regarding the undefined notations: As the details of the undefined notations were not specified, we assume you are referring to notations related to quantum mechanics. We have provided the necessary foundational information on quantum computing in the supplement, which covers all notations used in this paper. Please note that we assume some familiarity with quantum computing from our reviewers, as we cannot cover basic knowledge in this work.
> We have updated our work based on your suggestions, with the changes highlighted in red. We would greatly appreciate it if you could take a more detailed look at the paper and provide specific feedback on the proposed ideas, technical novelty, and experimental results, as well as further suggestions for improvement.

---

> > ### Comment · Reviewer_DCFN · 2024-11-25
> > **Comments on Authors' Rebuttal (1/3)**
> >
> > I have certainly carefully read your paper and I appreciate the effort that the authors made to revise the paper. However, I can not be more positive about this work. This decision is derived from the following concerns.
> >
> > (1) I do not think directly replacing some modules with a parameterized quantum circuit (PQC) can be viewed as a valid contribution after the developments of these years. Parameterized quantum circuits, due to their expressiveness, have been used to design hundreds of quantum machine learning algorithms in recent years [1]. This simple replacement should be viewed as a lack of both novelty and validity.
> >
> > Firstly for the novelty, the ansatz design (PQC used in this paper) does not show a significant difference from other related works [1, 2, 3, 4, 5, 6]. In QNF-Net, the design of quantum circuits is not novel, and only the way of extracting observable and post-process is explicitly designed. However, this process should be viewed as a classical computation instead of contributing as a quantum algorithm. Although Section 3.2 is entitled "PQC design", actually this is a design of how observable operators are chosen, which is kind of fundamental for almost all QML works based on PQC.
> >
> > Secondly for validity, without a special design of parameterized quantum circuits, the expressibility and the trainability are not guaranteed even theoretically, including the notorious barren plateaus problem [7]. Although a similar idea is mentioned in Section 3.2 (around lines 245-247), this significant drawback is ignored in the subsequent texts. Explicit quantum circuit design is not presented anywhere in this paper. A circuit with four qubits is shown in Figure 3, however, the figure seems to randomly put some quantum gate on the circuit. Importantly, recent works [7, 8, 9] show that specific circuit design could potentially solve this issue, without which quantum advantage will be eliminated [3, 7, 10, 11, 12].
> >
> > This paper discusses the expressibility in Section 3.1 (lemma 1) and Section 3.2 (theorem 1), showing that an arbitrary unitary matrix (line 226) W^1 could be regarded as a universal transformation, and can be decomposed by the famous Solovay-Kitaev theorem. However, I do not think it is appropriate to use the Solovay-Kitaev theorem here because it is unacceptable to have exponential growth of gate number. The usage of the theorem is not helpful in the context of PQC with polynomial circuit depth.

---

> > ### Comment · Reviewer_DCFN · 2024-11-25
> > **Comments on Authors' Rebuttal (2/3)**
> >
> > (2) The authors say that one of your main contributions is to use amplitude encoding to encode classical data. This is quite confusing from the quantum computing view for two reasons: Firstly, amplitude-encoding is extremely expensive for any NISQ quantum application, including most variational quantum algorithms. In fact, preparing an arbitrary amplitude-encoded state has been proven to be exponentially difficult [13, 14]. So I do not think this encoding method works in practice for a parameterized quantum circuit-based algorithm unless there are more efficient or insightful constructions. (Many PQC algorithms use other encoding methods such as angle-based encoding layers [3, 4]). In specific, this will result in a serious scalability problem [3, 7, 10, 11, 12]. Secondly, if your algorithm is for fault-tolerant applications instead of NISQ, then you should discuss more on potential quantum advantage. Otherwise, it will be really confusing to design a PQC algorithm with amplitude encoding in the fault-tolerant era.
> >
> > (3) The authors do analyze the circuit expressiveness. However, too universal expressiveness is, in some sense, a curse instead of a resource for quantum machine learning algorithms [3, 4, 7, 10, 11, 12]. Since it is very hard to train a PQC in a high-dimensional Hilbert space. Once again, this is relevant to the **trainability problem**. You do mention using initialization settings to mitigate this problem, however, this idea was proposed by other researchers and is not about QNF structure. In fact, your ansatz might have been shown to face barren plateaus in your numerical experiments, as reported in Fig. 8 of your work.
> >
> > Notably, it is not surprising that QNF-Net works in a small-scale numerical simulation[15, 16, 17, 18]. In fact, much more work should be done to show the scalability and the potential quantum advantage.
> >
> >
> >
> > To summarize, the distinct contribution of this work is insufficient and lacks novelty. QNF-Net is yet another direct substitution of some part in the classical algorithm by parametrized quantum circuit, with only empirical study on small-size quantum circuits. The common and core concerns from the quantum machine learning community have not been seriously regarded or properly addressed by this work, at least in its current version.
> >
> > The main motivation for introducing quantum computing into the area of machine learning is that we want to seek some advantages from quantum computing over classical computing. Using parameterized quantum circuit (or variational quantum circuit) in the encoding layer and trainable layer has been well-known in this area since 2017 [19], and thousands of subsequent works in these years[1, 3, 4, 5, 6, 8, 9, 10, 11, 12, 15, 16, 17, 18, 19]. Initially, people thought it was promising to have some advantage by this replacement. However, although extensive numerical experiments are conducted on small-scale settings (including simulations and real quantum computers), we can hardly find any convincing evidence of the quantum advantage. In recent years, many insightful theoretical works point out the limitations of quantum machine learning [3, 4, 7, 11, 12], and also some potential ways to overcome these issues, including special quantum circuit design [8, 9, 10, 20] and the quantum-specific tasks [21, 22]. These efforts tell us a substitution of a classical machine learning module to a quantum machine learning module requires carefully inspections on its motivations and potential quantum advantage. From this concern, I do not think this paper make a significant contribution to the quantum machine learning community.

---

> ### Comment · Reviewer_DCFN · 2024-11-25
> **Comments on Authors' Rebuttal (3/3)**
>
> References
>
> [1] Benedetti, Marcello, et al. "Parameterized quantum circuits as machine learning models." Quantum Science and Technology 4.4 (2019): 043001.
>
> [2]Sim, Sukin, Peter D. Johnson, and Alán Aspuru‐Guzik. "Expressibility and entangling capability of parameterized quantum circuits for hybrid quantum‐classical algorithms." Advanced Quantum Technologies 2.12 (2019): 1900070.
>
> [3]Cerezo, Marco, et al. "Challenges and opportunities in quantum machine learning." Nature Computational Science 2.9 (2022): 567-576.
>
> [4]Jeswal, S. K., and S. Chakraverty. "Recent developments and applications in quantum neural network: A review." Archives of Computational Methods in Engineering 26.4 (2019): 793-807.
>
> [5]Abbas, Amira, et al. "The power of quantum neural networks." Nature Computational Science 1.6 (2021): 403-409.
>
> [6]Beer, Kerstin, et al. "Training deep quantum neural networks." Nature communications 11.1 (2020): 808.
>
> [7]McClean, Jarrod R., et al. "Barren plateaus in quantum neural network training landscapes." Nature communications 9.1 (2018): 4812.
>
> [8]Das, Sreetama, and Filippo Caruso. "Permutation-equivariant quantum convolutional neural networks." Quantum Science and Technology (2024).
>
> [9]Liu, Feiyang, et al. "Information compression via hidden subgroup quantum autoencoders." npj Quantum Information 10.1 (2024): 74.
>
> [10]Larocca, Martín, et al. "Group-invariant quantum machine learning." PRX Quantum 3.3 (2022): 030341.
>
> [11]Perdomo-Ortiz, Alejandro, et al. "Opportunities and challenges for quantum-assisted machine learning in near-term quantum computers." Quantum Science and Technology 3.3 (2018): 030502.
>
> [12]Houssein, Essam H., et al. "Machine learning in the quantum realm: The state-of-the-art, challenges, and future vision." Expert Systems with Applications 194 (2022): 116512.
>
> [13] Zhang, Xiao-Ming, Tongyang Li, and Xiao Yuan. "Quantum state preparation with optimal circuit depth: Implementations and applications." Physical Review Letters 129.23 (2022): 230504.
>
> [14] Sun, Xiaoming, et al. "Asymptotically optimal circuit depth for quantum state preparation and general unitary synthesis." IEEE Transactions on Computer-Aided Design of Integrated Circuits and Systems 42.10 (2023): 3301-3314.
>
> [15] Cherrat, El Amine, et al. "Quantum vision transformers." Quantum 8.arXiv: 2209.08167 (2024): 1265.
>
> [16] Kerenidis, Iordanis, et al. "Quantum vision transformers." Quantum 8 (2024): 1265.
>
> [17] Unlu, Eyup B., et al. "Hybrid Quantum Vision Transformers for Event Classification in High Energy Physics." Axioms 13.3 (2024): 187.
>
> [18] Rajesh, Varadi, and Umesh Parameshwar Naik. "Quantum convolutional neural networks (QCNN) using deep learning for computer vision applications." 2021 International conference on recent trends on electronics, information, communication & technology (RTEICT). IEEE, 2021.
>
> [19]Biamonte, Jacob, et al. "Quantum machine learning." Nature 549.7671 (2017): 195-202.
>
> [20] Sauvage, Frederic, et al. "Building spatial symmetries into parameterized quantum circuits for faster training." Quantum Science and Technology 9.1 (2024): 015029.
>
> [21] Huang, Hsin-Yuan, et al. "Quantum advantage in learning from experiments." Science 376.6598 (2022): 1182-1186.
>
> [22] Ristè, Diego, et al. "Demonstration of quantum advantage in machine learning." npj Quantum Information 3.1 (2017): 16.

---

> ### Author Response · Authors · 2024-11-26
>
> Thank you for the interesting opinions, points and observations! Thank you for the possibility to discuss. Our work will definitely benefit from them, despite not fully agreeing with all of them as outlined in the following. It is great that different views on the related fields of research exist. We find in the detailed review even stronger evidence that our work will be a valuable contribution to ICLR and could ignite follow-up works. Along with that, we feel that rev. DCFN is somewhat generally sceptical towards QML and applies criteria of physics journals when evaluating our work (but not ML conferences). We believe this circumstance could bias the evaluation and should not justify paper rejection. As these differences in opinions concern some fundamental questions, they could serve as material for an open panel discussion with a respectful and open exchange. We are happy to address the raised concerns below. **Concern (2,3):**
> 1) First of all, our paper is not a theoretical machine learning paper (there might have been a misunderstanding). While large and fault-tolerant quantum computers are not there yet, it is understandable that a lot of work in the field is done relying on proofs (usual standard: worst-case assumptions). In contrast, we are doing things differently from the theoretical QML community. We rather target the general machine learning (ML) and computer vision (CV) communities. We follow the path of open exploration in contrast to the vast majority of QML works. We are motivated by addressing important and practical CV problems.
> Sometimes research follows the path from experiments to the theory: A working solution is demonstrated and afterwards, research is done to understand how and why it works from the theoretical perspective. Consider the stream engine invented long before the laws of thermodynamics have been formalised. Another prominent example in the context of classical techniques without guarantees is Simulated Annealing. It does not have provable guarantees for finding global optima. Nevertheless, it works remarkably well in practice.
> Moreover, consider the following evidence/argument: Classical state-of-the-art ML currently works beyond the provable regime. This could also be an argument in support of the experimental QML research. It is important to test different design choices. Experiments with QML models (also involving simulations) will very probably become increasingly more important in future. There is no final conclusion on QML yet, and there are many open research questions.
>
> 2) There is not only a single criterion for a paper to be valuable and relevant. We agree that not all theoretical QML researchers might necessarily find our work interesting (as reviewer DCFN is suggesting), which is fine and valid. Our contribution is addressing a new problem using QML. We demonstrate design choices that work empirically on a simulator of fault-tolerant quantum hardware, and we propose the first method addressing an important problem (esp. for the ML and CV communities) using the gate-based quantum computational paradigm. This alone makes the work valuable and worthy for these communities to become familiar with the results. This opinion—that empirical works are becoming increasingly important—is also shared by many researchers in the QML community.
> 3) Citing Rev. DCFN: “Much more work should be done to show the scalability and the potential advantage”. We are happy to hear that our work would have a strong potential to ignite follow-ups for further investigation. We find it as evidence of its relevance and timeliness. Demonstrating scalability and the potential advantage in the context of QNF would require not a single but a sequel of follow-up works. We believe the current submission reaches the level of strong technical innovation on the ICLR level. We also would like to encourage the ML and CV communities to think about QNFs and their drawbacks and help improve them. We are by no means claiming that all design choices are optimal. We are initiating a new research direction at the intersection of quantum computing, machine learning and computer vision with multiple possible follow-ups (as DCFN agrees with us).
>
> 4) Regarding NISQ and the sensitivity of amplitude encoding to noise: We assume fault-tolerant quantum computers to address the fundamental challenges of designing a neural field with PQCs. We have adjusted the beginning of Sec. 3 accordingly. We strongly believe it is a valid assumption that is worth investigating. A follow-up could investigate the influence of the noise occurring in NISQ devices but first, the results for fault-tolerant machines have to be known and understood.

---

> ### Author Response · Authors · 2024-11-26
>
> 5) “...with only empirical study on small-size quantum circuits.” While our circuits are moderate in size, we are able to represent 2D images and 3D shapes of resolutions previously unseen in the literature in the context of QML models. The scale of these experiments goes beyond the existing state of the art and provides fundamental evidence of the validity of our method and the design choices.
> While much of the current theoretical research in QML addresses small-scale problems, its applicability to real-world and large-scale scenarios remains unclear. This gap underscores the importance of designing quantum or quantum-inspired models that can handle real-world scenarios while demonstrating tangible advantages (i.e., faster convergence or a smaller number of parameters). With our work successfully reconstructing dense 3D fields for the first time, we believe it holds experimental significance and makes a meaningful contribution.
> 6) The high-level question of whether quantum machine learning (QML) genuinely offers advantages compared to the classical method is an open question in the field that would require many more years of research by multiple research groups. While we provide a tiny (on the scale of the entire field) but important and promising experimental evidence supporting the scalability of QML models, we believe answering the general question regarding QML is beyond our scope and the scope of any single paper). Both experimental and theoretical QML research is required to find the answers in the long term, as many assumptions in theoretical research are often invalid in practical settings.
> 7) Regarding the trainability issue: This work is not a study focused on trainability analysis or other open challenges in theoretical QML. Our primary objective is to develop a neural field representation model for practical applications storing data in the transformation parameters of the PQCs. While we acknowledge that barren plateaus in quantum circuits are often associated with expressivity and measures like Haar randomness, our integration of classical modules to prepare such circuits introduces additional complexity, and the trainability of this hybrid approach has not been fully explored. Nevertheless, in our experiments—aimed at designing quantum circuits capable of scaling to real-world datasets, such as those used in 3D scene reconstruction—we achieve promising results with competitive performance compared to classical (but far not toy or outdated) baselines. Note that many published papers in QML do not reach this level of experimental comparisons. Further theoretical investigations into trainability or related issues fall beyond the scope of this work but could be considered in future.
> 8) Amplitude encoding (AE). AE is chosen in this submission because it is a widely adopted technique across various applications and has been demonstrated to be effective in numerous contexts including 3D. While it is true that angle encoding could be easier to implement in real-world physical systems, it remains unclear how to use it efficiently to reach arbitrary quantum states. Additionally, there has been significant progress in developing efficient amplitude encoding methods; see, e.g., Ashhab et al., Phys. Rev. Research 4, 013091 (2022); Conde et al., Quantum 8, 1297 (2024); and Daimon et al., "Quantum Circuit Generation for Amplitude Encoding Using a Transformer Decoder," Physical Review Applied 22.4 (2024).

---

> ### Author Response · Authors · 2024-11-26
>
> **Concern 1:**
> We have thoroughly reviewed the provided references. Regarding the novelty of our PQC design, we deliberately restrict quantum processing to the Bloch sphere around the Y-axis within our quantum circuit design. This design choice is supported by both theoretical analysis and empirical validation and preserves performance while significantly simplifying the optimization process. These insights are particularly impactful at the hardware level, where more complex quantum operations are prone to noise and lack efficient implementation methods. Regarding observable extraction, we believe that it is a valid quantum operation as it entails measuring a quantum system in an appropriate basis to extract meaningful information.
> We now address the concerns about the validity of our approach. Our work focuses on designing a model that leverages quantum circuits to learn neural field representations at the large scale shown in the literature for the first time while demonstrating an improvement to classical baselines. Although we provide some theoretical analysis of our model (see lines 215–230), its robustness is validated comprehensively through our experimental results (see Section 4 and visualized outcomes). Our quantum circuit is designed following the fully entangled pattern.
> Regarding the mitigation of the barren plateaus (BPs): we incorporated several techniques, including optimized initialization strategies and localized measurements, to mitigate BPs. Since our final model successfully learns large-scale field representations, we did not explore this problem further, as our primary emphasis is on the empirical design and observations, as stated at the outset of the paper.
> Lastly, regarding the citation of the Solovay-Kitaev theorem: We emphasize that this theorem is not invoked here to justify exponential gate growth. Instead, it establishes the theoretical guarantee that any unitary operation can be approximated to arbitrary precision using a universal gate set. This foundational principle also contextualizes the observed experimental performance improvements with deeper quantum circuits.
>
> Conclusion. Given all that, we are hence kindly requesting the reviewer to adjust and rethink the evaluation criteria (or/and the confidence level), as we feel the reviewer applies too strong criteria and presumably criteria of physics journals when evaluating our work. Both theoretical and experimental QML enrich each other and can benefit from synergies. We believe that readers should decide the future of this work at this point, and we hope ICLR will allow us to present it and respectfully and openly exchange opinions.

---

> ### Comment · Reviewer_DCFN · 2024-11-27
>
> I am delighted to put all the problems on the table. I have the following suggestions.
>
> 1. Directly using amplitude encoding requires more details and checks.
>
> (1) There is no explicit way to encode arbitrary data into amplitude encoding, unless you use QRAM or some other simplifications.
> "We can then prepare our final quantum encoding...Eq. (4)" -  No we can't. (This is partially why I said "missing details")
> The authors should indicate this in the paper with appropriate discussions.
>
> (2) The description of Eq. (5) is wrong. First, when g(x)= exp(-iPx) where P is a Pauli operator, f(x) is a sine-wave instead of a Fourier sum. Second, I guess the authors want to say g(x) represents the encoding for all data, where g(x)=g_k(x) g_{k-1}(x) ... g_1(x), and each g_i(x) is a single-qubit rotation. However, this paper uses the amplitude encoding - this cannot derive Eq. (5).
>
> Eq. (5) and Schuld et al. 2021 are not compatible with Eq. (4) and amplitude encoding. So it is questionable for the statement of "multi-dimensional frequency spectrum" as well as the "expressiveness of the model", negatively affecting the soundness of the theoretical part of the paper.
>
> A minor issue: In Eq. (4), what is |\psi_i\rangle? (Typically this is the computational basis)
>
> 2. The complexity of the algorithm on a quantum computer should be discussed.
>
> 3. Using Theorem 1 to imply the "universal expressiveness" is inappropriate. The expressiveness should only be discussed when the circuit is of polynomial depth. Theorem 1 is only a textbook-level theorem that shows any unitary can be decomposed of a number of elementary gates, which is unrelated to this paper.

---

> > ### Author Response · Authors · 2024-11-29
> >
> > **concern 1.1:**
> >
> > We agree that there has not reached an agreement on how to perform a universal and efficient amplitude encoding (AE) for arbitrary classical data (e.g., encoded into normalised probability distributions). Along with that, the research field of physical AE realisations remains active/is progressing (point 8 on AE of the previous response round): Ashhab et al., Phys. Rev. Research 4, 013091 (2022); Conde et al., Quantum 8, 1297 (2024); and Daimon et al., "Quantum Circuit Generation for Amplitude Encoding Using a Transformer Decoder," Phys. Rev. Applied 22.4 (2024). Tackling both high- and low-level aspects in a single work is not feasible, as both require dedicated attention and full paper formats. We did not want to exclude AE from the range of potential design choices for QNF-Net as we consider it is a very promising encoding strategy of classical data which is widely adopted as the default encoding method in many quantum algorithms, despite not all aspects of its efficient implementation being yet widely explored.
> >
> > We adjusted the Discussion and Conclusion section to highlight that physical realisations of AE are an important and open aspect when attempting to implement QNF-Net on real quantum hardware (repeated here for convenience):
> >
> > *While a standard, widely accepted approach for efficient amplitude encoding of arbitrary classical normalized data has yet to emerge, the field of its physical realizations is advancing at a remarkable pace; see Ashhab et al., Phys. Rev. Research 4, 013091 (2022); Conde et al., Quantum 8, 1297 (2024); and Daimon et al., "Quantum Circuit Generation for Amplitude Encoding Using a Transformer Decoder," Physical Review Applied 22.4 (2024). The proposed innovative association between amplitude encoding and energy could potentially inspire the preparation of such quantum states with Hamiltonian evolution using devices such as quantum annealers. Furthermore, future research could explore alternative data encoding strategies that could become more practical as the implementation of QNF-Net on real quantum hardware becomes feasible. Even in the absence of suitable quantum hardware in the near term, our QNF remains valid as a quantum-inspired method. While it does not yet account for all aspects of hardware realization, it offers valuable insights and progresses the field. Lastly, beyond the demonstrated applications in shape interpolation and completion, QNF also shows promise for tasks such as image and shape classification.*
> >
> > Another consideration supporting the current method version with AE is that, as hinted by Rev. ZVMN, QNF-Net offers advantages w.r.t. classical techniques already now, i.e., when implemented on a simulator of quantum hardware. Certainly, QNF-Net could also be called a “quantum-inspired approach” until more aspects of physical realisations (of AE and other components) are clarified once suitable quantum computers are available.
> >
> > The proposed encoding strategy (classical ML + AE that associates energy with the corresponding probability distribution prior rather than arbitrarily normalized data), has the potential to inspire the preparation of the encoded states using devices such as quantum annealers, which are inherent quantum Boltzmann samplers, that leverage specific Hamiltonian evolutions.

---

> ### Author Response · Authors · 2024-11-29
>
> **concern 1.2:**
>
> We believe Eq. (5) is correct in the current form after a careful examination upon the reviewer’s request. We agree that its interpretation and connection to AE can be improved. Hence, we updated the corresponding paper section.
> While Eq. (5) originally applies to angle encoding explicitly, i.e. g(x)= exp(-iPx), angle embedding can approximate amplitude encoding by iteratively encoding data in the angles of quantum gates and leveraging the expressive power of parameterized quantum circuits. The theoretical basis for this approximation can be partially explained through the Solovay-Kitaev theorem, stating that any unitary operation can be approximated to arbitrary precision using a finite set of universal gates given sufficient circuit depth. In this context, angle embedding can be seen as constructing an approximation of the transformations required for AE by composing parameterized rotation gates. With the increasing circuit depth, the fidelity of this approximation improves, allowing angle embedding to effectively simulate the state preparation required for AE. In this work, we use Eq.(5) as a general representation of the circuit structure that is an ansatz–encoding–ansatz framework. The encoding module g(x) in Eq. (5) is versatile and covers a wide range of encoding strategies (it is not limited to  Pauli encoding, i.e. angular encoding with Pauli gates, and can include multiple data-reuploading operations).
>
> The discussion of the connection between AE and angle embedding also highlights their inherent links to the Fourier domain. In this context, the multi-dimensional spatial field is first abstracted into an energy field representation in a learnable manner. Consequently, the Fourier spectrum becomes intrinsically related to this abstracted energy field and is specifically determined by the eigenvalues of the quantum circuit gates required to encode the data. This relationship allows us to naturally associate the inferred energy levels with the multi-dimensional frequency spectra, as detailed in ll. 218–230.
>
> **concern 2:**
>
> In Eq.4, |\psi_i\rangle refers to a general and local basis. Notably, the computational basis serves as a valid example of such a basis to represent our prepared quantum state. We have updated the draft accordingly (see L214 in the latest updated draft).
>
> Analyzing the computational complexity of executing quantum circuits on physical hardware is a challenging task due to various factors. We are hesitant we can provide a precise estimate in this work. Gate fidelity, hardware architecture, types of gates, and the specifics of physical implementation all play critical roles. Fidelity and noise levels directly affect accuracy and may necessitate additional error correction, while hardware constraints like qubit connectivity can increase the required circuit depth. Furthermore, different quantum platforms, such as superconducting qubits or trapped ions, come with unique requirements for gate execution speed and reliability. These combined factors make assessing the practical complexity of running quantum circuits on hardware a non-trivial problem.
>
> While it is uncommon to discuss computational complexity for practical QML work, we try our best to address your question from an algorithmic perspective. The complexity of running a parametrized quantum circuit can be estimated by the number of gates in the circuit (assuming physical factors are ignored). In our method design, we optimized the circuit design and partially constrained its expressivity without compromising performance with given theoretical evidence and empirical verification. This approach reduces the number of gates required compared to other QML works that fully exploit rotations across the entire Bloch sphere.
>
> **concern 3:**
>
> We agree and do not claim that the Solovay-Kitaev theorem directly establishes expressiveness in circuits of polynomial depth. We reference it to highlight the general principle that any desired unitary transformation can be approximated, albeit with circuit depth bounded—at worst—by an exponential factor. This also accounts for the experimental observation that with increasing circuit depth, the performance improves due to the circuit’s ability to approximate the target measurements with greater precision. We would like to emphasize that, as this is an ML conference, we assume some of our readers are less familiar with certain quantum computing principles. Therefore, we included and explained some of the relevant concepts in the draft.

---

### Official Review · Reviewer_e8T4 · 2024-11-02

**Soundness:** 3
**Presentation:** 4
**Contribution:** 4
**Rating:** 8
**Confidence:** 3

**Summary:**

The paper provides a novel approach to implicit neural representations by encoding scenes using parameterised quantum circuits. The method involves a coordinate and latent code encoding through a lightweight MLP (classical component) to generate a energy manifold, which is used to construct a quantum circuit (quantum component). Outputs are generated by sampling from the circuit. Experiments are run on a quantum simulator on image regression and occupancy field experiments, showing higher accuracy and improved convergence relative to baseline MLP models. The authors provide detailed theoretical descriptions of the quantum circuit and methodology, and perform a number of ablations to support their findings.

**Strengths:**

The main strength of the paper is in its novel approach for combining QML with implicit neural representations. Aside from one contemporaneous paper (QIREN Zhao et al. 2024 https://arxiv.org/abs/2406.03873, which seems to use a very different approach; and a preprint Quantum Radiance Fields, Yuan-Fu and Sun 2023 https://arxiv.org/abs/2211.03418), this appears to be one of the only papers on this topic. This is therefore significant as it extends a QML methodology to the important and rapidly expanding area of implicit neural representation research.

The paper is clearly written and provides sufficient background details for both INR and QML literature to act as a great bridging paper for researchers in each area. Theoretical details are clearly described, and presentation in general is at a very high level both for written descriptions and figures (e.g. Figures 1, 2, and 4). Experiments are conducted on a variety of implicit neural representations (images and occupancy fields; image in-painting and shape completion), with appropriate metrics showing improvements relative to baselines. Ablations (qubits, block repetitions, encoder layers, and periodic activations) are appropriate and show a thorough evaluation.

**Weaknesses:**

While the paper presents a novel, well-described, and interesting approach to implicit neural representation training, there are a few weaknesses of the paper relating to experimental conclusions (e.g. parameter improvements compared to traditional INR approaches).

Image experiments are conducted on CIFAR10 images. Very small MLPs can be used to fit these and larger images to high quality (see: Dupont et al. 2021 https://arxiv.org/abs/2103.03123, who fit larger Kodak images [768, 512, 3] with smaller networks of 2,000 - 15,000 parameters). This indicates that the classical energy encoding component of the network (15,000 parameters) may already be over-specified with respect to the target signal and doing the heavy lifting relative to the quantum parameters. Figure 8 shows relatively little difference with the number of qubits used (although it is highly sensitive to the number of classical layers, and partially sensitive to the number of block repetitions). This again calls into question whether the classical or quantum component doing more of the processing of the signal. It would be useful if the authors could discuss this and check two baselines: 1) A more restricted (in terms of parameters) classical energy encoder and MLP baseline; 2) To evaluate the quantum parameters (e.g. distribution of values at initialisation vs following training, to check whether these exhibit much change).

**Questions:**

The authors restrict their circuits to use only real-valued components to simplify the optimization problem. Could the authors provide any details about the impact of loosening this constraint? In addition, it would be useful to discuss in more detail the limitations of practically applying this method on a quantum computer rather than a simulator (e.g. backpropagation through the quantum circuit as described in Appendix A).

The authors note that they encounter memory depletion issues due to storage of intermediate results (L375). Could the authors possibly describe this issue (e.g. what are the memory requirements of the method for the image / occupancy experiments, time required for convergence, etc)?

---

> ### Author Response · Authors · 2024-11-23
>
> Thank you for your insightful and valuable comments! We are encouraged by your acknowledgement of our technical novelty in combining QML with neural implicit representations and its significance to the rapidly growing implicit neural representation research. We would also like to thank you for appraising the clarity of our writing and paper structure. Below, we carefully address your questions and concerns.
>
> **Impact of loosening the constraint:**
>
> We agree that relaxing the constraint would be an interesting direction, which we have identified as potential future work. One approach could involve replacing the hard constraint with a soft constraint implemented through penalization. We plan to explore this in our future research
>
> **Running on real gate-based quantum computers:**
>
> We also agree that running quantum circuits on real quantum computers is an exciting avenue. However, despite recent advancements in circuit-based quantum hardware, access to a sufficient number of qubits with the required stability and quality remains highly restricted. Moreover, our attempt to use a real circuit-based quantum computer provided by IBM was hindered by unrealistic queuing time.
>
> Given our focus on exploring the potential of these models and their applications to large-scale datasets, we have opted to use high-end simulators as an alternative. While we acknowledge the importance of conducting experiments on real hardware, the current constraints necessitate simulation on classical computers even for those work that use small dataset; see Rathi et al. "3D-QAE: Fully Quantum Auto-Encoding of 3D Point Clouds." (2023); Baek et al. "3D scalable quantum convolutional neural networks for point cloud data processing in classification applications." arXiv preprint arXiv:2210.09728 (2022); Zhao et al. "Quantum Implicit Neural Representations." Forty-first International Conference on Machine Learning.
>
> **Memory depletion issues:**
>
> Regarding memory depletion, we encountered issues when extending our experiments to dense 3D reconstructions. Specifically, for 3D scenes with a sampling density of 500k points per scene (the original setting where we faced memory issues), we were unable to train the model on our A100 GPU with 80 GB of memory. In terms of training time, the model takes approximately 3 hours for 2D image data and 8 hours for 3D scene data, though this also depends on factors such as sampling density, learning rate, model size or even the type of simulator used.

---

> ### Author Response · Authors · 2024-11-24
>
> **Concern: Quantum circuit learning and its significance:**
>
> Thank you for your detailed observation and for highlighting that smaller MLPs can serve as effective alternatives for fitting CIFAR-10 images, raising important questions about the significance of each model component. Since our experiments span both 2D images and dense 3D scene reconstructions, we carefully scaled our networks to ensure consistent parameter counts across tasks, providing clarity and comparability. While smaller parameter counts may suffice for CIFAR-10, 3D scene learning typically demands significantly more parameters; see Park et al. "Deepsdf: Learning continuous signed distance functions for shape representation." . As shown in Table 1, incorporating a medium-scaled quantum circuit under strictly controlled conditions enabled us to achieve better performance using only 70% of the original parameters. Since the quantum circuit was the only variable introduced, we believe this validates its effectiveness in enhancing the overall model. Additionally, we fully agree that visualizing the evolution of parameters in the quantum circuit is insightful and therefore have included such a figure in the appendix of our revised draft, with updates highlighted in red.

---

> > ### Comment · Reviewer_e8T4 · 2024-11-25
> >
> > I thank the authors for their rebuttal and have reviewed the revised draft. I suggest that the authors use the previous version of Table 1 (without the confidence interval), and include the confidence interval information visually (e.g. bar charts) in the supplementary materials. The updated table is very difficult to parse in contrast to the original.
> >
> > Could the authors additionally please clarify the specific architectures used for experiments in Table 1 and 2 (hidden layers, units per layer, etc) for the MLP baseline, SIREN, classical encoder, and quantum circuit as requested by reviewer DCFN.

---

> > > ### Author Response · Authors · 2024-11-26
> > >
> > > We thank the reviewer for the suggestions to improve clarity and readability. In response, we revert to the previous table version and will include the confidence interval information in the supplementary material, as recommended. We also agree and recognize the importance of providing architectural details in the main matter. For consistency of the experiments, our baseline MLP employs three hidden layers, with 328 neurons each in the baseline configuration. Next, we begin with a  small number of neurons per layer for the energy inference module and gradually increase its scale until sufficient to achieve superior results; the scale is about 70 percent of its original, corresponding to 256 neurons per layer. In the Siren configuration, we preserve the parameter count while replacing the classical ReLU with periodic activation, following the initialization strategy described in the original Siren paper.
> > > In our quantum circuit, we employ a fully entangled pattern and constrain the rotations to the Y-axis (rather than allowing full Bloch sphere rotations). Both theoretical analysis and empirical results indicate that this design choice does not degrade the performance and significantly simplifies the optimization process. We believe such design considerations are essential for advancing practical QML applications.

---

> ### Comment · Reviewer_e8T4 · 2024-11-27
>
> I thank the authors for providing more detail about the model architectures, and suggest including these encoder ablations in the supplementary materials for completeness. While I do have reservations about the baseline comparisons, I believe the main strength of this paper is in its novel application of QML techniques to INRs, which will be of interest and worth accepting to ICLR.

---

> ### Author Response · Authors · 2024-12-02
>
> We greatly appreciate your suggestion regarding encoder ablations. We launched additional systematic experiments and will include the results in the supplementary material. Thank you once again for your feedback and recognition of the novelty of our work in applying QML techniques to INRs for the first time.

---

### Official Review · Reviewer_ZVMN · 2024-11-03

**Soundness:** 2
**Presentation:** 2
**Contribution:** 3
**Rating:** 5
**Confidence:** 3

**Summary:**

This paper introduces QNF-Net, a hybrid quantum-classical learning framework designed for visual computing tasks. QNF-Net combines an innovative neuro-deterministic encoding module, which maps classical data into quantum states via an energy inference process, with a parameterized quantum circuit (PQC) for efficient visual data modeling. Through experimental results on 2D images and 3D shapes, the authors show that QNF-Net outperforms classical MLP baselines in both accuracy and convergence speed, indicating potential quantum advantages in neural field representation.

**Strengths:**

The authors propose an original framework of applying quantum machine learning for neural field representations, with a novel approach for input data encoding. The experiments highlight the potential of QNF-Net for faster convergence and higher parameter efficiency relative to traditional MLPs, suggesting promising applications for quantum-enhanced neural fields. The results provide a compelling indication that quantum computing could benefit data-intensive fields, such as visual computing, where efficiency and scalability are critical.

**Weaknesses:**

The experimental section could benefit from further depth, particularly in baseline comparisons and scalability analysis. The chosen MLP baseline, while helpful for initial comparisons, lacks contextualization with more recent advancements in neural field representations.

This limitation also weakens the argument for quantum advantage, especially given that QNF-Net is only tested with up to 6 qubits, a relatively small-scale quantum setup, and it is not clear why more qubits leads to worse performance. Additionally, the paper would be strengthened by discussing potential quantum hardware issues, such as noise and decoherence, as these are critical for real-world applicability.

Finally, Sec. 4.4 on alternative energy inference designs feels disconnected from the core narrative and could be integrated more cohesively into the broader experimental results.

**Questions:**

1.	The proposed encoding method shares some similarities with neural network quantum states (e.g., Carleo and Troyer, Science 355, 602 (2017)), where a similar energy function is used to parameterize quantum states. Could the authors clarify the conceptual similarities and differences between these approaches?
2.	In Eq. (3), the normalization appears to only account for spatial dimensions but not the quantum state’s spin degrees of freedom. For a system with n qubits, summing over 2^n basis states would become infeasible as n grows. How is the normalization handled in practice for large n?
3.	Could the authors elaborate on how the proposed data encoding would be implemented on a physical quantum computer? Details on how the Gibbs-Boltzmann energy distribution could be prepared as qubit amplitudes would enhance understanding.
4.	What causes the spikes in the loss curves in Fig. 8?
5.	Fig. 1(c) feels out of place and is not referenced until Section 4, which impacts flow and clarity. Additionally, Figs. 1, 2, and 3 contain overlapping information. A reorganization could help improve clarity and readability.

---

> ### Author Response · Authors · 2024-11-23
>
> Thank you for your insightful and valuable comments. We are encouraged by your acknowledgement of our motivation, technical soundness, experimental results, and recognition of our work as a compelling indication that quantum computing can benefit data-intensive fields such as visual computing. Below, we carefully address your questions and concerns.
>
> **question 1:**
>
> Similarities:
>
> Both works draw inspiration from the universal approximation capabilities of neural networks, leveraging them to tackle problems across different domains. Carleo and Troyer, Science (2017) instead of numerically evaluating the interaction of quantum many-body systems and simulating the evolution of wave functions with time, propose an approach to learning many-body wave functions and their evolution.
>
> Differences:
>
> Our work introduces latent vectors for conditional learning, enabling simultaneous training on diverse data and generating unseen samples via latent space interpolations, positioning our approach as a generative network—a direction unexplored by Carleo et al. Unlike their focus on using AI to simulate quantum wavefunction evolution without encoding classical data as quantum states, we enhance classical machine learning with quantum computing through a novel learning-based amplitude encoding, linking classical data to energy inference with neural networks. Additionally, we propose optimized quantum circuit modules with a simplified parameter search space, improving efficiency without sacrificing performance. Targeting field representation in visual computing, our model extends the solvable problem scale to reconstruct multiple high-quality 3D shapes, contrasting with Carleo et al.'s focus on many-body wavefunctions, reflecting fundamentally distinct data processing needs.
>
> **question 2:**
>
> In our experiment under the specified setting, we observed that increasing the number of qubits provides limited performance improvement. However, as the system scales with an increasing number of qubits n, summing over an exponentially growing number of terms quickly becomes computationally infeasible. A practical solution to this challenge is to encourage sparsity during learning through regularization and to prune values below a predefined small threshold.
>
> **question 3:**
>
> While our energy inference scheme is executed on classical hardware (see the method figure), it is entirely feasible to prepare the quantum states on quantum hardware. For example, quantum states with amplitudes following a Gibbs-Boltzmann energy distribution can be prepared on a quantum annealer by implementing a specific Hamiltonian evolution schedule, leveraging the annealer's inherent function as a quantum Boltzmann machine (Amin et al., Phys. Rev. X 8, 021050 (2018)). Moreover, as the proposed energy-based encoding aligns with the broader category of amplitude encoding, various implementation techniques have been explored in the literature (e.g., Conde et al., Quantum 8, 1297 (2024); Ashhab et al., Phys. Rev. Research 4, 013091 (2022)).
>
> **question 4:**
>
> The spikes observed in the training curve during our experiments may be attributed to the coordination challenges between the classical and quantum modules. Since the energies and features generated and inferred by the classical network module and the quantum circuit have inherently different interpretations, we think it is reasonable to observe such behavior.
>
> **question 5:**
>
> We thank the reviewer for pointing this out. We will update the figures/tables/sections for better readability in the revised version.

---

> ### Author Response · Authors · 2024-11-24
>
> **Concern 1: Baseline comparison**
>
> With quantum machine learning still in its infancy, MLP baselines remain a robust choice and have yet to be surpassed in many CV/CG studies; see Rathi et al. in "3D-QAE: Fully Quantum Auto-Encoding of 3D Point Clouds." Our model enhances fully connected neural networks with quantum circuits, offering fundamentally different theoretical expressivity. Consequently, we chose to benchmark it against a similarly structured network with enhanced performance—Siren, an MLP architecture that leverages periodic activation functions for greater robustness compared to standard ReLU. In our experiments, we demonstrated that our model not only achieved better metric performance and used fewer parameters than standard MLPs but also outperformed Siren, a comparison that, to our knowledge, has not yet been explored in quantum machine learning research. While many recent advancements in neural field representations, such as Mildenhall et al.'s NeRF, Chen et al.'s MobileNeRF, and Shue et al.'s 3D Neural Field Generation Using Triplane Diffusion, focus on optimizing applications and efficiency, they rely on architectures beyond fully connected networks and often involve massive parameter counts. For these reasons, we did not consider them for comparison in our work.
>
> **Concern 2: Quantum circuit scale**
>
> We acknowledge your concerns, however, simulating deep quantum circuits with many qubits is computationally intensive, especially for large-scale datasets. In this work, as we focus on developing a model capable of handling large-scale datasets, such as dense 3D shape reconstructions, we need to consider minimizing quantum resource usage while preserving quantum advantage.
>
> **Concern 3: Quantum hardware issues such as noise and decoherence:**
>
> As there has not been a model capable of scaling up to 3D reconstruction before our work, the primary goal of this study is to develop a model that can handle large datasets while maintaining quantum advantage, assuming the availability of noise-tolerant quantum computers. Therefore, we did not place significant emphasis on the effect of noise or its potential influence on our experiments. However, in the presence of noise, we would anticipate substantial performance degradation, as shown by other works that primarily investigate the impact of noise on machine learning; see Borras et al., Journal of Physics: Conference Series.Vol.2438.No.1.IOP Publishing, 2023.
>
> **Concern 4: Structural layout**
>
> We thank you for pointing it out. We will work to incorporate it into the broader experimental results as suggested.

---

> > ### Comment · Reviewer_ZVMN · 2024-11-25
> >
> > I thank the authors for the detailed explanations. While most of my questions and concerns have been addressed, one key question remains: why quantum? From my understanding, the quantum advantage does not scale up even if more qubits are available, and this will likely get worse on realistic quantum hardware.
> >
> > From this perspective, the proposed framework could be better positioned as a quantum-inspired algorithm, where classical simulations of small quantum systems yield a significant performance boost. However, it is unclear why this algorithm would need to be executed on actual quantum hardware.
> >
> > I would be willing to increase my score if the authors can provide convincing evidence or arguments that this concern is unfounded.

---

> > > ### Author Response · Authors · 2024-12-02
> > >
> > > **Additional thoughts**: We would like to add that our primary focus is to demonstrate the potential advantages of integrating quantum circuits or quantum-inspired operations into neural architectures, for answering fundamental research questions associated with this idea as well as for scaled-up applications (w.r.t. previous QML works) like 3D shape completion—a topic rarely explored in the context of QML. For medium-scale fields with coarser resolutions, GPU-accelerated simulators (a standard tool for evaluating quantum advantages prior to quantum hardware deployment in the future) have already shown promising results. Hence, we agree our approach can be considered quantum-inspired. In the long term, representing fields with higher resolutions is expected to benefit from execution on real quantum hardware, where the extended capabilities of quantum computation (w.r.t. classical hardware) can be additionally (and, potentially, fully) harnessed.

---

> ### Author Response · Authors · 2024-11-26
>
> We thank you for asking this question and we agree it is important to clarify it. The observation that the current model works and can be executed on a simulator of a fault-tolerant quantum computer is correct. Quantum gate sets such as the Clifford allow for efficient classical simulation. Unfortunately, most QML models rely on gates outside these efficiently simulatable sets, rendering their classical simulation computationally expensive. This highlights the potential for circuit-based quantum computers to offer significant advantages in future. Next, even with more qubits involved in our experiment and stricter control of other conditions, we did not observe significant improvements in model convergence, at least in the tested settings. The scalability of a quantum circuit depends not only on the number of qubits but also on the circuit depth and the types of gates employed—both of which directly influence execution runtime on classical simulators. Currently, we prioritize the depth of the PQC instead of the qubit count in our design. On real quantum hardware, one can in future expect scaling QNF-Net to a higher number of qubits (>20) thus further increasing the supported data sample resolution and even faster training and convergence compared to a classical baseline. Note that the model with more qubits currently trains slower but supports a higher resolution of the learned signals (2D-pixel grids and sampled surfaces for SDFs).

---

### Author Response · Authors · 2024-11-23

We sincerely thank all reviewers for their valuable feedback. We are encouraged by the acknowledgement of our work's strengths and novelty, including our novel policy for data encoding and its compelling direction (Reviewer ZVMN), a unique method for implicit neural representations (Reviewers e8T4, ZVMN),  a new alternative for representing neural fields (Reviewer McH9), and a interesting model for CV/CG (Reviewer DCFN). In our individual responses, we provide detailed explanations for the raised concerns.

---

### Meta-Review · Area_Chair_Qwj6 · 2024-12-20

**Metareview:**

The submission introduces QNF-Net, a quantum neural field network designed for visual computing tasks, including 2D image representation and 3D shape reconstruction. The proposed approach integrates neuro-deterministic encoding with PQCs, enabling the mapping of classical data into quantum states. Experimental results on quantum simulators demonstrate the method's potential effectiveness. While the submission presents intriguing ideas that combine QML with neural field representations, it falls short in convincingly addressing the meaningfulness and practical utility of the proposed approach in its current form.

**Additional Comments On Reviewer Discussion:**

During the discussion period, the primary issues centered on concerns regarding the practical utility of the proposed QNF-Net, the scalability of the method to larger datasets and quantum circuits, and the feasibility of implementing it on real quantum hardware. While the authors provided clarifications and additional context, significant concerns about the practical utility of QNF-Net remained. As highlighted by reviewers ZVMN and DCFN, the submission lacked concrete evidence to substantiate the indispensability of quantum computers in advancing neural field representations. Although this work represents an initial attempt in this direction, these unresolved issues may limit its impact within the QML community.

---

### Decision · Program_Chairs · 2025-01-22

Reject